# CURRICULUM LOSS: ROBUST LEARNING AND GENERALIZATION AGAINST LABEL CORRUPTION

**Yueming Lyu & Ivor W. Tsang**
Centre for Artificial Intelligence, University of Technology Sydney
`yueminglyu@gmail.com, Ivor.Tsang@uts.edu.au`

## ABSTRACT

Deep neural networks (DNNs) have great expressive power, which can even memorize samples with wrong labels. It is vitally important to reiterate robustness and generalization in DNNs against label corruption. To this end, this paper studies the 0-1 loss, which has a monotonic relationship with empirical adversary (reweighted) risk (Hu et al., 2018). Although the 0-1 loss has some robust properties, it is difficult to optimize. To efficiently optimize the 0-1 loss while keeping its robust properties, we propose a very simple and efficient loss, i.e. curriculum loss (CL). Our CL is a tighter upper bound of the 0-1 loss compared with conventional summation based surrogate losses. Moreover, CL can adaptively select samples for model training. As a result, our loss can be deemed as a novel perspective of curriculum sample selection strategy, which bridges a connection between curriculum learning and robust learning. Experimental results on benchmark datasets validate the robustness of the proposed loss.

## 1 INTRODUCTION

Noise corruption is a common phenomenon in our daily life. For instance, noisy corrupted (wrong) labels may be resulted from annotating for similar objects (Su et al., 2012; Yuan et al., 2019), crawling images and labels from websites (Hu et al., 2017; Tanaka et al., 2018) and creating training sets by program (Ratner et al., 2016; Khetan et al., 2018). Learning with noisy labels is thus an promising area.

Deep neural networks (DNNs) have great expressive power (model complexity) to learn challenging tasks. However, DNNs also undertake a higher risk of overfitting to the data. Although many regularization techniques, such as adding regularization terms, data augmentation, weight decay, dropout and batch normalization, have been proposed, generalization is still vitally important for deep learning to fully exploit the super-expressive power. Zhang et al. (2017) show that DNNs can even fully memorize samples with incorrectly corrupted labels. Such label corruption significantly degenerates the generalization performance of deep models. This calls a lot of attention on robustness in deep learning with noisy labels.

**Robustness of 0-1 loss:** The problem resulted from data corruption or label corruption is that test distribution is different from training distribution. Hu et al. (2018) analyzed the adversarial risk that the test distribution density is adversarially changed within a limited $f$-divergence (e.g. KL-divergence) from the training distribution density. They show that there is a monotonic relationship between the (empirical) risk and the (empirical) adversarial risk when the 0-1 loss function is used. This suggests that minimizing the empirical risk with the 0-1 loss function is equivalent to minimize the empirical adversarial risk (worst-case risk). When we train a model based on the corrupted training distribution, we want our model to perform well on the clean distribution. Since we do not know the clean distribution, we want our model to perform well for the worst case estimate of the clean distribution in some constrained set. It is thus natural to employ the worst-case classification risk of the estimated clean distribution as the objective. Note that the worst-case classification risk is an upper bound of the classification risk of the true clean distribution, minimizing the worst-case risk can usually decrease the true risk. When we employ the 0-1 loss, because of the equivalence between the classification risk and the worst-case classification risk, we can directly minimize the

classification risk under the corrupted training distribution instead of minimizing the worst-case classification risk.

From the learning perspective, the 0-1 loss is more robust to outliers compared with an unbounded (convex) loss (e.g. hinge loss) (Masnadi-Shirazi & Vasconcelos, 2009). This is due to unbounded convex losses putting much weight on the outliers (with a large loss value) when minimizing the losses (Masnadi-Shirazi & Vasconcelos, 2009). If the unbounded (convex) loss is employed in deep network models, this becomes more prominent. Since training loss of deep networks can often be minimized to zero, outlier with a large loss has a large impact on the model. On the other hand, the 0-1 loss treats each training sample equally. Thus, each sample does not have too much influence on the model. Therefore, the model is tolerant of a small number of outliers.

Although the 0-1 loss has many robust properties, its non-differentiability and zero gradients make it difficult to optimize. One possible way to alleviate this problem is to seek an upper bound of the 0-1 loss that is still efficient to optimize but tighter than conventional (convex) losses. Such a tighter upper bound of the 0-1 loss can reduce the influence of the noisy outliers compared with conventional (convex) losses. At the same time, it is easier to optimize compared with the 0-1 loss. When minimizing the upper bound surrogate, we expect that the 0-1 loss objective is also minimized.

**Learnability under large noise rate:** The 0-1 loss cannot deal with large noise rate. When the noise rate becomes large, the systematic error (due to label corruption) grows up and becomes not negligible. As a result, the model's generalization performance will degenerate due to this systematic error. To reduce the systematic error produced by training with noisy labels, several methods have been proposed. They can be categorized into three kinds: transition matrix based method (Sukhbaatar et al., 2014; Patrini et al., 2017; Goldberger & Ben-Reuven, 2017), regularization based method (Miyato et al., 2016) and sample selection based method (Jiang et al., 2018; Han et al., 2018b). Among them, sample selection based method is one promising direction that selects samples to reduce noisy ratio for training. These methods are based on the idea of curriculum learning (Bengio et al., 2009) which is one successful method that trains the model gradually with samples ordered in a meaningful sequence. Although they achieve success to some extents, most of these methods are heuristic based.

To efficiently minimize the 0-1 loss while keeping the robust properties, we propose a novel loss that is a tighter upper bound of the 0-1 loss compared with conventional surrogate losses. Specifically, giving any base loss function $l(u) \geq \mathbf{1}(u < 0), u \in \mathbb{R}$, our loss $Q(\mathbf{u})$ satisfies $\sum_{i=1}^{n} \mathbf{1}(u_i < 0) \leq Q(\mathbf{u}) \leq \sum_{i=1}^{n} l(u_i)$, where $\mathbf{u} = [u_1, \cdots, u_n]$ with $u_i$ being the classification margin of $i^{th}$ sample, and $\mathbf{1}(\cdot)$ is an indicator function. We name it as Curriculum Loss (CL) because our loss automatically and adaptively selects samples for training, which can be deemed as a curriculum learning paradigm.

Our contributions are listed as follows:

- We propose a novel loss (i.e. curriculum loss) for robust learning against label corruption. We prove that our CL is a tighter upper bound of 0-1 loss compared with conventional summation based surrogate loss. Moreover, CL can adaptively select samples for stagewise training, which bridges a connection between curriculum learning and robust learning.
- We prove that CL can be performed by a simple and fast selection algorithm with $\mathcal{O}(n \log n)$ time complexity. Moreover, our CL supports mini-batch update, which is convenient to be used as a plug-in in many deep models.
- We further propose a Noise Pruned Curriculum Loss (NPCL) to address label corruption problem by extending CL to a more general form. Our NPCL automatically prune the estimated noisy samples during training. Moreover, NPCL is also very simple and efficient, which can be used as a plug-in in deep models as well.

## 2 CURRICULUM LOSS

In this section, we present the framework of our proposed Curriculum Loss (CL). We begin with discussion about robustness of the 0-1 loss in Section 2.1. We then show that our CL is a tighter upper bound of the 0-1 loss compared with conventional summation based surrogate losses in Section 2.2. A tighter bound of the 0-1 loss means that it is less sensitive to the noisy outliers, and it

better preserves the robustness of the 0-1 loss with a small rate of label corruption. For a large rate of label corruption, we extend our CL to a Noise Pruned Curriculum Loss (NPCL) to address this issue in Section 2.3. A simple multi-class extension and a novel soft multi-hinge loss are included in the Appendix. All the detailed proofs can be found in the Appendix as well.

## 2.1 ROBUSTNESS OF 0-1 LOSS AGAINST LABEL CORRUPTION

We rephrase Theorem 1 in (Hu et al., 2018) from a different perspective, which motivates us to employ the 0-1 loss for training against label corruption.

**Theorem 1.** *(Monotonic Relationship) (Hu et al. (Hu et al., 2018)) Let $p(x, y)$ and $q(x, y)$ be the training and test density, respectively. Define $r(x, y) = q(x, y)/p(x, y)$ and $r_i = r(x_i, y_i)$. Let $l(\widehat{y}, y) = \mathbf{1}\big(sign(\widehat{y}) \neq y\big)$ and $l(\widehat{y}, y) = \mathbf{1}\big(argmax_k(\widehat{y}_k) \neq y\big)$ be 0-1 loss for binary classification and multi-class classification, respectively. Let $f(\cdot)$ be convex with $f(1) = 0$. Define risk $\mathcal{R}(\theta)$, empirical risk $\widehat{\mathcal{R}}(\theta)$, adversarial risk $\mathcal{R}_{adv}(\theta)$ and empirical adversarial risk $\widehat{\mathcal{R}}_{adv}(\theta)$ as*

$$\mathcal{R}(\theta) = \mathbb{E}_{p(x,y)}\left[l(g_\theta(x), y)\right] \tag{1}$$

$$\widehat{\mathcal{R}}(\theta) = \frac{1}{n}\sum\nolimits_{i=1}^{n} l(g_\theta(x_i), y_i) \tag{2}$$

$$\mathcal{R}_{adv}(\theta) = \sup_{r \in \mathcal{U}_f} \mathbb{E}_{p(x,y)}\left[r(x, y)l(g_\theta(x), y)\right] \tag{3}$$

$$\widehat{\mathcal{R}}_{adv}(\theta) = \sup_{\mathbf{r} \in \widehat{\mathcal{U}}_f} \frac{1}{n}\sum\nolimits_{i=1}^{n} r_i l(g_\theta(x_i), y_i), \tag{4}$$

*where $\mathcal{U}_f = \big\{r(x, y) \big| \mathbb{E}_{p(x,y)}\left[f\left(r(x, y)\right)\right] \leq \delta, \mathbb{E}_{p(x,y)}\left[r(x, y)\right] = 1, r(x, y) \geq 0, \forall(x, y) \in \mathcal{X} \times \mathcal{Y}\big\}$ and $\widehat{\mathcal{U}}_f = \big\{\mathbf{r} \big| \frac{1}{n}\sum_{i=1}^{n} f(r_i) \leq \delta, \frac{1}{n}\sum_{i=1}^{n} r_i = 1, \mathbf{r} \geq 0\big\}$. Then we have that*

$$\textit{If } \mathcal{R}_{adv}(\theta_1) < 1, \quad \textit{then } \mathcal{R}(\theta_1) < \mathcal{R}(\theta_2) \iff \mathcal{R}_{adv}(\theta_1) < \mathcal{R}_{adv}(\theta_2). \tag{5}$$

$$\textit{If } \mathcal{R}_{adv}(\theta_1) = 1, \quad \textit{then } \mathcal{R}(\theta_1) \leq \mathcal{R}(\theta_2) \iff \mathcal{R}_{adv}(\theta_2) = 1. \tag{6}$$

*The same monotonic relationship holds between their empirical approximation: $\widehat{\mathcal{R}}(\theta)$ and $\widehat{\mathcal{R}}_{adv}$.*

Theorem 1 (Hu et al., 2018) shows that the monotonic relationship between the (empirical) risk and the (empirical) adversarial risk (worst-case risk) when 0-1 loss function is used. It means that minimizing (empirical) risk is equivalent to minimize the (empirical) adversarial risk (worst-case risk) for 0-1 loss. When we train a model based on the corrupted training distribution $p(x, y)$, we want our model to perform well on the clean distribution $q(x, y)$. Since we do not know the clean distribution $q$, we want our model to perform well for the worst-case estimate of the clean distribution, with the assumption that the $f$-divergence between the corrupted distribution $p$ and the clean distribution $q$ is bounded by $\delta$. Note that the underlying clean distribution is fixed but unknown, given the corrupted training distribution, the smallest $\delta$ that bounds the divergence between the corrupted distribution and clean distribution measures the intrinsic difficulty of the corruption, and it is also fixed and unknown. The corresponding worst-case distribution w.r.t the smallest $\delta$ is an estimate of the true clean distribution, and this worst-case risk upper bounds the risk of the true clean distribution. In addition, this bound is tighter than the other worst-case risks w.r.t larger $\delta$. It is natural to use this upper bound as the objective for robust learning. When we use 0-1 loss (that is commonly employed for evaluation), because of the equivalence of the risk and the worst-case risk, we can directly minimize risk under training distribution $p$ instead of directly minimizing the worst-case risk (i.e., the upper bound). Moreover, this enables us to minimize the upper bound without knowing the true $\delta$ beforehand. When the true $\delta$ is small, i.e., the corruption of the training data is not heavy, the upper bound is not too pessimistic. Usually, minimizing the upper bound can decrease the true risk under clean distribution. Particularly, when the clean distribution coincides with the worst-case estimate w.r.t the smallest $\delta$, minimizing the risk under the corrupted training distribution leads to the same minimizer as minimizing the risk under the clean distribution.

## 2.2 TIGHTER UPPER BOUNDS OF THE 0-1 LOSS

Unlike commonly used loss functions in machine learning, the non-differentiability and zero gradients of the 0-1 loss make it difficult to optimize. We thus propose a tighter upper bound surrogate

loss. We use the classification margin to define the 0-1 loss. For binary classification, classification margin is $u = \hat{y}y$, where $\hat{y}$ and $y \in \{+1, -1\}$ denotes the prediction and ground truth, respectively. (A simple multi-class extension is discussed in the Appendix.) Let $u_i \in \mathbb{R}$ be the classification margin of the $i^{th}$ sample for $i \in \{1, ..., n\}$. Denote $\mathbf{u} = [u_1, ..., u_n]$. The 0-1 loss objective can be defined as follows:

$$J(\mathbf{u}) = \sum_{i=1}^{n} \mathbf{1}(u_i < 0). \tag{7}$$

Given a base upper bound function $l(u) \geq \mathbf{1}(u < 0), u \in \mathbb{R}$, the conventional surrogate of the 0-1 loss can be defined as

$$\widehat{J}(\mathbf{u}) = \sum_{i=1}^{n} l(u_i). \tag{8}$$

Our curriculum loss $Q(\mathbf{u})$ can be defined as Eq.(9). $Q(\mathbf{u})$ is a tighter upper bound of 0-1 loss $J(\mathbf{u})$ compared with the conventional surrogate loss $\widehat{J}(\mathbf{u})$, which is summarized in Theorem 2:

**Theorem 2.** *(Tighter Bound) Suppose that base loss function $l(u) \geq \mathbf{1}(u < 0), u \in \mathbb{R}$ is an upper bound of the 0-1 loss function. Let $u_i \in \mathbb{R}$ be the classification margin of the $i^{th}$ sample for $i \in \{1, ..., n\}$. Denote $\max(\cdot, \cdot)$ as the maximum between two inputs. Let $\mathbf{u} = [u_1, ..., u_n]$. Define $Q(\mathbf{u})$ as follows:*

$$Q(\mathbf{u}) = \min_{\mathbf{v} \in \{0,1\}^n} \max \left( \sum_{i=1}^{n} v_i l(u_i), n - \sum_{i=1}^{n} v_i + \sum_{i=1}^{n} \mathbf{1}(u_i < 0) \right). \tag{9}$$

*Then $J(\mathbf{u}) \leq Q(\mathbf{u}) \leq \widehat{J}(\mathbf{u})$ holds true.*

**Remark:** For any fixed $\mathbf{u}$, we can obtain an optimum solution $\mathbf{v}^*$ of the partial optimization. The index indicator $\mathbf{v}^*$ can naturally select samples as a curriculum paradigm for training models. The partial optimization w.r.t index indicator $\mathbf{v}$ can be solved by a very simple and efficient algorithm (Algorithm 1) in $\mathcal{O}(n \log n)$. Thus, the loss is very efficient to compute. Moreover, since $Q(\mathbf{u})$ is tighter than conventional surrogate loss $\widehat{J}(\mathbf{u})$, it is less sensitive to outliers compared with $\widehat{J}(\mathbf{u})$. Furthermore, it better preserves the robust property of the 0-1 loss against label corruption.

The difficulty of optimizing the 0-1 loss is that the 0-1 loss has zero gradients in almost everywhere (except at the breaking point). This issue prevents us from using first-order methods to optimize the 0-1 loss. Eq.(9) provides a surrogate of the 0-1 loss with non-zero subgradient for optimization, while preserving robust properties of the 0-1 loss. Note that our goal is to construct a tight upper bound of the 0-1 loss while maintaining informative (sub)gradients. Eq.(9) balances the 0-1 loss and conventional surrogate by selecting (the trust) samples (index) for training progressively.

Updating with all the samples at once is not efficient for deep models, while training with mini-batch is more efficient and well supported for many deep learning tools. We thus propose a batch based curriculum loss $\widehat{Q}(\mathbf{u})$ given as Eq.(10). We show that $\widehat{Q}(\mathbf{u})$ is also a tighter upper bound of 0-1 loss objective $J(\mathbf{u})$ compared with conventional loss $\widehat{J}(\mathbf{u})$. This property is summarized in Corollary 1.

**Corollary 1.** *(Mini-batch Update) Suppose that base loss function $l(u) \geq \mathbf{1}(u < 0), u \in \mathbb{R}$ is an upper bound of the 0-1 loss function. Let $b, m$ be the number of batches and batch size, respectively. Let $u_{ij} \in \mathbb{R}$ be the classification margin of the $i^{th}$ sample in batch $j$ for $i \in \{1, ..., m\}$ and $j \in \{1, ..., b\}$. Denote $\mathbf{u} = [u_{11}, ..., u_{mb}]$. Let $n = mb$. Define $\widehat{Q}(\mathbf{u})$ as follows:*

$$\widehat{Q}(\mathbf{u}) = \sum_{j=1}^{b} \min_{\mathbf{v} \in \{0,1\}^m} \max \left( \sum_{i=1}^{m} v_{ij} l(u_{ij}), m - \sum_{i=1}^{m} v_{ij} + \sum_{i=1}^{m} \mathbf{1}(u_{ij} < 0) \right). \tag{10}$$

*Then $J(\mathbf{u}) \leq Q(\mathbf{u}) \leq \widehat{Q}(\mathbf{u}) \leq \widehat{J}(\mathbf{u})$ holds true.*

**Remark:** Corollary 1 shows that a batch-based curriculum loss is also a tighter upper bound of 0-1 loss $J(\mathbf{u})$ compared with the conventional surrogate loss $\widehat{J}(\mathbf{u})$. This enables us to train deep models with mini-batch update. Note that random shuffle in different epoch results in a different batch-based curriculum loss. Nevertheless, we at least know that all the induced losses are upper bounds of 0-1 loss objective and are tighter than $\widehat{J}(\mathbf{u})$. Moreover, all these losses are induced by the same base loss function $l(\cdot)$. Note that, our goal is to minimize the 0-1 loss. Random shuffle leads to a multiple surrogate training scheme. In addition, training deep models without shuffle does not have this issue.

We now present another curriculum loss $E(\mathbf{u})$ which is tighter than $Q(\mathbf{u})$. $E(\mathbf{u})$ is an (scaled) upper bound of 0-1 loss. This property is summarized as Theorem 3.

---

**Algorithm 1** Partial Optimization

---

**Input:** $u_i$ for $i \in \{1, ..., n\}$, the selection threshold $C$;
**Output:** Index set $\mathbf{v} = (v_1, v_2, \ldots, v_n)$;
Compute the losses $l_i = l(u_i)$ for $i = 1, ..., n$;
Sort samples (index) w.r.t. the losses $\{l_i\}_{i=1}^n$ in a non-decreasing order;      // Get $l_1 \leq \cdots \leq l_n$
Initialize $L_0 = 0$;
**for** $i = 1$ **to** $n$ **do**
   $L_i = L_{i-1} + l_i$;
   **if** $L_i \leq (C + 1 - i)$ **then**
     Set $v_i = 1$;
   **else**
     Set $v_i = 0$;
   **end if**
**end for**

---

**Theorem 3.** *(Scaled Bound) Suppose that base loss function $l(u) \geq \mathbf{1}(u < 0), u \in \mathbb{R}$ is an upper bound of the 0-1 loss function. Let $u_i \in \mathbb{R}$ be the classification margin of the $i^{th}$ sample for $i \in \{1, ..., n\}$. Denote $\mathbf{u} = [u_1, ..., u_n]$. Define $E(\mathbf{u})$ as follows:*

$$E(\mathbf{u}) = \min_{\mathbf{v} \in \{0,1\}^n} \max \big( \sum\nolimits_{i=1}^n v_i l(u_i), n - \sum\nolimits_{i=1}^n v_i \big). \tag{11}$$

*Then $J(\mathbf{u}) \leq 2E(\mathbf{u}) \leq 2\widehat{J}(\mathbf{u})$ holds true.*

**Remark:** $E(\mathbf{u})$ has similar properties to $Q(\mathbf{u})$ discussed above. Moreover, it is tighter than $Q(\mathbf{u})$, i.e. $E(\mathbf{u}) \leq Q(\mathbf{u})$. Thus, it is less sensitive to outliers compared with $Q(\mathbf{u})$. However, $Q(\mathbf{u})$ can construct more adaptive curriculum by taking 0-1 loss into consideration during the training process.

Directly optimizing $E(\mathbf{u})$ is not as efficient as that optimizing $Q(\mathbf{u})$. We now present a batch loss objective $\widehat{E}(\mathbf{u})$ given as Eq.(12). $\widehat{E}(\mathbf{u})$ is also a tighter upper bound of 0-1 loss objective $J(\mathbf{u})$ compared with conventional surrogate loss $\widehat{J}(\mathbf{u})$.

**Corollary 2.** *(Mini-batch Update for Scaled Bound) Suppose that base loss function $l(u) \geq \mathbf{1}(u < 0), u \in \mathbb{R}$ is an upper bound of the 0-1 loss function. Let $b$, $m$ be the number of batches and batch size, respectively. Let $u_{ij} \in \mathbb{R}$ be the classification margin of the $i^{th}$ sample in batch $j$ for $i \in \{1, ..., m\}$ and $j \in \{1, ..., b\}$. Denote $\mathbf{u} = [u_{11}, ..., u_{mb}]$. Let $n = mb$. Define $\widehat{E}(\mathbf{u})$ as follows:*

$$\widehat{E}(\mathbf{u}) = \sum\nolimits_{j=1}^b \min_{\mathbf{v} \in \{0,1\}^m} \max \big( \sum\nolimits_{i=1}^m v_{ij} l(u_{ij}), m - \sum\nolimits_{i=1}^m v_{ij} \big). \tag{12}$$

*Then $J(\mathbf{u}) \leq 2E(\mathbf{u}) \leq 2\widehat{E}(\mathbf{u}) \leq 2\widehat{J}(\mathbf{u})$ holds true.*

All the curriculum losses defined above rely on minimizing a partial optimization problem (Eq.(13)) to find the selection index set $\mathbf{v}^*$. We now show that the optimization of $\mathbf{v}$ with given classification margin $u_i \in \mathbb{R}, i \in \{1, ..., n\}$ can be done in $\mathcal{O}(n \log n)$.

**Theorem 4.** *(Partial Optimization) Suppose that base loss function $l(u) \geq \mathbf{1}(u < 0), u \in \mathbb{R}$ is an upper bound of the 0-1 loss function. For fixed $u_i \in \mathbb{R}$, $i \in \{1, ..., n\}$, an minimum solution $\mathbf{v}^*$ of the minimization problem in Eq. (13) can be achieved by Algorithm 1:*

$$\min_{\mathbf{v} \in \{0,1\}^n} \max \big( \sum\nolimits_{i=1}^n v_i l(u_i), C - \sum\nolimits_{i=1}^n v_i \big), \tag{13}$$

*where $C$ is the threshold parameter such that $0 \leq C \leq 2n$.*

**Remark:** The time complexity of Algorithm 1 is $\mathcal{O}(n \log n)$. Moreover, it does not involve complex operations, and is very simple and efficient to compute.

Algorithm 1 can adaptively select samples for training. It has some useful properties to help us better understand the objective after partial minimization, we present them in Proposition 1.

**Proposition 1.** *(Optimum of Partial Optimization) Suppose that base loss function $l(u) \geq \mathbf{1}(u < 0), u \in \mathbb{R}$ is an upper bound of the 0-1 loss function. Let $u_i \in \mathbb{R}$ for $i \in \{1, ..., n\}$ be fixed values. Without loss of generality, assume $l(u_1) \leq l(u_2) \cdots \leq l(u_n)$. Let $\mathbf{v}^*$ be an optimum solution of the partial optimization problem in Eq.(13). Let $T^* = \sum_{i=1}^{n} v_i^*$ and $L_{T^*} = \sum_{i=1}^{T^*} l(u_i)$. Then we have*

$$L_{T^*} \leq C + 1 - T^* \tag{14}$$

$$L_{T^*+1} > C - T^* \tag{15}$$

$$L_{T^*+1} > \max(L_{T^*}, C - T^*) \tag{16}$$

$$\min_{\mathbf{v} \in \{0,1\}^n} \max \big( \sum_{i=1}^{n} v_i l(u_i), C - \sum_{i=1}^{n} v_i \big) = \max(L_{T^*}, C - T^*). \tag{17}$$

**Remark:** When $C \leq n + \sum_{i=1}^{n} \mathbf{1}(u_i < 0)$, Eq.(17) is tighter than the conventional loss $\widehat{J}(\mathbf{u})$. When $C \geq n$, Eq. (17) is a scaled upper bound of 0-1 loss $J(\mathbf{u})$. From Eq.(17), we know the optimum of the partial optimization problem (13) (i.e. our objective) is $\max(L_{T^*}, C - T^*)$. When $L_{T^*} \geq C - T^*$, we can directly optimize $L_{T^*}$ with the selected samples for training. When $L_{T^*} < C - T^*$, note that $L_{T^*+1} > \max(L_{T^*}, C - T^*)$ from Eq.(16), we can optimize $L_{T^*+1}$ for training. Note that when $T^* < n$, we have that $L_{T^*+1} \leq L_n = \sum_{i=1}^{n} l(u_i)$, which is still tighter than the conventional loss $\widehat{J}(\mathbf{u})$. When $T^* = n$, for the parameter $C \leq n + \sum_{i=1}^{n} \mathbf{1}(u_i < 0)$, we have that $L_{T^*} = \widehat{J}(\mathbf{u}) \geq J(\mathbf{u}) \geq C - n = C - T^*$. Thus we can optimize $\max(L_{T^*}, C - T^*) = \widehat{J}(\mathbf{u})$. In practice, when training with random mini-batch, we find that optimizing $L_{T^*}$ in both cases instead of $L_{T^*+1}$ does not make much influence.

## 2.3 NOISE PRUNED CURRICULUM LOSS

The curriculum loss in Eq.(9) and Eq.(11) expect to minimize the upper bound of the 0-1 loss for all the training samples. When model capability (complexity) is high, (deep network) model will still attain small (zero) training loss and overfit to the noisy samples.

The ideal model is that it correctly classifies the clean training samples and misclassifies the noisy samples with wrong labels. Suppose that the rate of noisy samples (by label corruption) is $\epsilon \in [0, 1]$. The ideal model is to correctly classify the $(1 - \epsilon)n$ clean training samples, and misclassify the $\epsilon n$ noisy training samples. This is because the label is corrupted. Correctly classify the training samples with corrupted (wrong) label means that the model has already overfitted to noisy samples. This will harm the generalization to the unseen data.

Considering all the above reasons, we thus propose the Noise Pruned Curriculum Loss (NPCL) as

$$\mathcal{L}(\mathbf{u}) = \min_{\mathbf{v} \in \{0,1\}^n} \max \big( \sum_{i=1}^{n} v_i l(u_i), C - \sum_{i=1}^{n} v_i \big), \tag{18}$$

where $C = (1 - \epsilon)n$ or $C = (1 - \epsilon)^2 n + (1 - \epsilon) \sum_{i=1}^{n} \mathbf{1}(u_i < 0)$.

When we know there are $\epsilon n$ noisy samples in the training set, we can leverage this as our prior. (The impact of misspecification of the prior is included in the supplement.) When $C = (1 - \epsilon)n$ (assume $C, \epsilon n$ are integers for simplicity), from the selection procedure in Algorithm 1, we know $\epsilon n$[1] samples with largest losses $l(u)$ will be pruned. This is because $C - \sum_{i=1}^{n} v_i + 1 \leq 0$ when $\sum_{i=1}^{n} v_i \geq (1 - \epsilon)n + 1$. Without loss of generality, assume $l(u_1) \leq l(u_2) \cdots \leq l(u_n)$. After pruning, we have $v_{(1-\epsilon)n+1} = \cdots = v_n = 0$, the pruned loss becomes

$$\widetilde{\mathcal{L}}(\mathbf{u}) = \min_{\mathbf{v} \in \{0,1\}^{(1-\epsilon)n}} \max \big( \sum_{i=1}^{(1-\epsilon)n} v_i l(u_i), (1 - \epsilon)n - \sum_{i=1}^{(1-\epsilon)n} v_i \big). \tag{19}$$

It is the basic CL for $(1 - \epsilon)n$ samples and it is the upper bound of $\sum_{i=1}^{(1-\epsilon)n} \mathbf{1}(u_i < 0)$. If we prune more noisy samples than clean samples, it will reduce the noise ratio. Then the basic CL can handle. Fortunately, this assumption is supported by the "memorization" effect in deep networks (Arpit et al., 2017), i.e. deep networks tend to learn clean and easy pattern first. Thus, the loss of noisy or hard

---

[1]When $\sum_{i=1}^{(1-\epsilon)n+1} l(u_i) \neq 0$, $\epsilon n$ samples will be pruned. Otherwise, $\epsilon n - 1$ samples will be pruned.

---

**Algorithm 2** Training with Batch Noise Pruned Curriculum Loss

---

**Input:** Number of epochs $N$, batch size $m$, noise ratio $\epsilon$;
**Output:** The model parameter $\mathbf{w}$;
Initialize model parameter $\mathbf{w}$.
**for** $k = 1$ **to** $N$ **do**
    Shuffle training set $\mathcal{D}$;
    **while** Not fetch all the data from $\mathcal{D}$ **do**
        Fetch a mini-batch $\widehat{\mathcal{D}}$ from $\mathcal{D}$;
        Compute losses $\{l_i\}_{i=1}^m$ for data in $\widehat{\mathcal{D}}$;
        Compute the selection threshold $C$ according to Eq.(21).
        Compute selection index $\mathbf{v}^*$ by Algorithm 1;
        Update $\mathbf{w} = \mathbf{w} - \alpha \nabla l\left(\widehat{\mathcal{D}}_{\mathbf{v}^*}\right)$ w.r.t the subset $\widehat{\mathcal{D}}_{\mathbf{v}^*}$ of $\widehat{\mathcal{D}}$ selected by $\mathbf{v}^*$;
    **end while**
**end for**

---

data tend to remain high for a period (before being overfitted). Therefore, the pruned samples with largest loss are more likely to be the noisy samples. After the rough pruning, the problem becomes optimizing basic CL for the remaining samples as in Eq.(19). Note that our CL is a tight upper bound approximation to the 0-1 loss, it preserves the robust property to some extent. Thus, it can handle case with small noise rate. Specifically, our CL(Eq.19) further select samples from the remaining samples for training adaptively according to the state of training process. This generally will further reduce the noise ratio. Thus, we may expect our NPCL to be robust to noisy samples. Note that, all the above can be done by the simple and efficient Algorithm 1 without explicit pruning samples in a separated step. Namely, our loss can do all these automatically under a unified objective form in Eq.(18).

When $C = (1 - \epsilon)n$, the NPCL in Eq.(18) reduces to basic CL $E(\mathbf{u})$ in Eq.(11) with $\epsilon = 0$. When $C = (1-\epsilon)^2 n + (1-\epsilon)\sum_{i=1}^n \mathbf{1}\left(u_i < 0\right)$, for an ideal target model (that misclassifies noisy samples only), we know that $\mathbb{E}[C] = (1 - \epsilon)^2 n + (1 - \epsilon)\mathbb{E}[\sum_{i=1}^n \mathbf{1}\left(u_i < 0\right)] = (1 - \epsilon)^2 n + (1 - \epsilon)\epsilon n = (1 - \epsilon)n$. It has similar properties as choosing $C = (1 - \epsilon)n$. Moreover, it is more adaptive by considering 0-1 loss during training at different stages. In this case, the NPCL in Eq.(18) reduces to the CL $Q(\mathbf{u})$ in Eq.(9) when $\epsilon = 0$. Note that $C$ is a prior, users can defined it based on their domain knowledge.

To leverage the benefit of deep learning, we present the batched NPCL as

$$\widehat{\mathcal{L}}(\mathbf{u}) = \sum_{j=1}^b \min_{\mathbf{v} \in \{0,1\}^m} \max\left(\sum_{i=1}^m v_{ij} l(u_{ij}), \widehat{C}_j - \sum_{i=1}^m v_{ij}\right), \quad (20)$$

where $\widehat{C}_j = (1 - \epsilon)m$ or as in Eq.(21):

$$\widehat{C}_j = (1 - \epsilon)^2 m + (1 - \epsilon)\sum_{i=1}^m \mathbf{1}\left(u_{ij} < 0\right). \quad (21)$$

Similar to Corollary 1, we know that $\mathcal{L}(\mathbf{u}) \leq \widehat{\mathcal{L}}(\mathbf{u})$. Thus, optimizing the batched NPCL is indeed minimizing the upper bound of NPCL. This enables us to train the model with mini-batch update, which is very efficient for modern deep learning tools. The training procedure is summarized in Algorithm 2. It uses Algorithm 1 to select a subset of samples from every mini-batch. Then, it uses the selected samples to perform gradient update.

## 3 EMPIRICAL STUDY

### 3.1 EVALUATION OF ROBUSTNESS AGAINST LABEL CORRUPTION

We evaluate our NPCL by comparing Generalized Cross-Entropy (GCE) loss (Zhang & Sabuncu, 2018), Co-teaching (Han et al., 2018b), Co-teaching+ (Yu et al., 2019), MentorNet (Jiang et al., 2018) and standard network training on MNIST, CIFAR10 and CIFAR100 dataset as in (Han et al., 2018b; Patrini et al., 2017; Goldberger & Ben-Reuven, 2017). Two types of random label corruption, i.e. Symmetry flipping (Van Rooyen et al., 2015) and Pair flipping (Han et al., 2018a), are

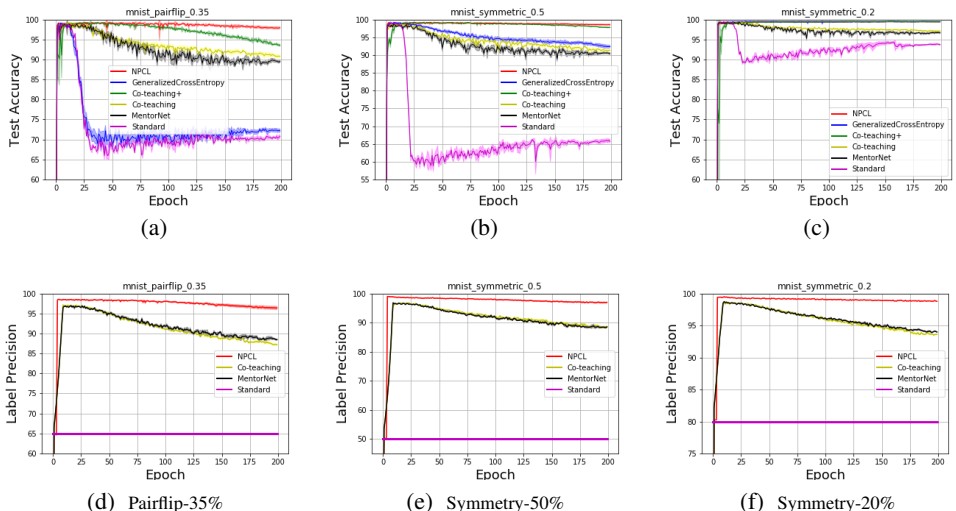

Figure 1: Test accuracy and label precision vs. number of epochs on MNIST dataset.

considered in this work. Symmetry flipping is that the corrupted label is uniformly assign to one of $K-1$ incorrect classes. Pair flipping is that the corrupted label is assign to one specific class similar to the ground truth. The noise rate $\epsilon$ of label flipping is chosen from $\{20\%, 50\%, 35\%\}$ as a representative. As a robust loss function, we further compare NPCL with GCE loss in detail with noise rate in $\{0\%, 10\%, 20\%, 30\%, 40\%, 50\%\}$. We employ same network architecture and network hyperparameters as in Co-teaching (Han et al., 2018b) for all the methods in comparison. Specifically, the batch size and the number of epochs is set to $m = 128$ and $N = 200$, respectively. The Adam optimizer with the same parameter as (Han et al., 2018b) is employed. The architecture of neural network is presented in Appendix L. For NPCL, we employ hinge loss as the base upper bound function of 0-1 loss. In the first few epochs, we train model using full batch with soft hinge loss (in the supplement) as a burn-in period suggested in (Jiang et al., 2018). Specifically, we start NPCL at $5^{th}$ epoch on MNIST and $10^{th}$ epoch on CIFAR10 and CIFAR100, respectively. For Co-teaching (Han et al., 2018b) and MentorNet in (Jiang et al., 2018), we employ the open sourced code of Co-teaching (Han et al., 2018b). For Co-teaching+ (Yu et al., 2019), we employ the code provided by the authors. We implement NPCL by Pytorch. For NPCL, Co-teaching and Co-teaching+, we employ the true noise rate as parameter. Experiments are performed five independent runs. The error bar for STD is shaded.

For performance measurements, we employ both test accuracy and label precision as in (Han et al., 2018b). Label precision is defined as : *number of clean samples / number of selected samples*, which measures the selection accuracy for sample selection based methods. A higher label precision in the mini-batch after sample selection can lead to a update with less noisy samples, which means that model suffers less influence of noisy samples and thus preforms more robustly to label corruption.

The pictures of test accuracy and label precision vs. number of epochs on MNIST are presented in Figure 1. The results on CIFAR10 and CIFAR100 are shown in Figure 5 and Figure 6 in Appendix, respectively. It shows that NCPL achieves superior performance compared with GCE loss in terms of test accuracy. Particularly, NPCL obtains significant better performance compared with GCE loss in hard cases: Symmetry-50% and Pair-flip-35%, which shows that NPCL is more robust to label corruption compared with GCE loss. Moreover, NPCL obtains better performance on MNIST, and competitive performance on CIFAR10 and CIFAR100 compared with Co-teaching. Furthermore, NPCL achieves better performance than Co-teaching+ on CIFAR10 and two cases on MNIST. In addition, we find that Co-teaching+ is not stable on CIFAR100 with 50% symmetric noise. Note that NPCL is a simple plug-in for a single network, while Co-teaching/Co-teaching+ employs two networks to train the model concurrently. Thus, both the space complexity and time complexity of Co-teaching/Co-teaching+ is doubled compared with our NPCL.

Both our NPCL and Generalized Cross Entropy (GCE) loss are robust loss functions as plug-in for single network. Thus, we provide a more detailed comparison between our NPCL and GCE loss with noise rate in $\{0\%, 10\%, 20\%, 30\%, 40\%, 50\%\}$. The experimental results on CIFAR10 are presented in Figure 3. The experimental results on CIFAR100 and MNIST are provided in Figure 8 and Figure 7 in Appendix.From Figure 3, Figure 8 and Figure 7, we can observe that NPCL obtains similar and higher test accuracy in all the cases. Moreover, from Figure 3 and Figure 7, we can see that NPCL achieves similar test accuracy compared with the GCE loss when the noise rate is small. The improvement increases with the increase of the noise rate. Particularly, NPCL obtains remarkable improvement compared with the GCE loss on CIFAR10 with noise rate 50%. It shows that NPCL is more robust compared with GCE loss against label corruption. GCE loss employs all samples for training, while NPCL prunes the noisy samples adaptively. As a result, GCE loss still employs samples with wrong labels for training, which misleads the model. Thus, NPCL obtains better performance when the noise rate becomes large.

## 3.2 MORE EXPERIMENTS WITH DIFFERENT NETWORK ARCHITECTURES

We follow the experiments setup in (Lee et al., 2019). We use the online code of (Lee et al., 2019) , and only change the loss for comparison. We cite the numbers of Softmax, RoG and D2L (Ma et al., 2018) in (Lee et al., 2019) for comparison.

The test accuracy results on uniform noise, semantic noise and open-set noise are shown in Table 1, Table 2 and Table 3, respectively. From Table 1, we can observe that both NPCL and CL outperforms Softmax (cross-entropy) and RoG (cross-entropy) on five cases for uniform noise. Note that RoG is an ensemble method, while CL/NPCL is a single loss for network training, one can combine them to boost the performance. From Table 2, we can see that CL obtains consistently better performance than cross-entropy and D2L (Ma et al., 2018) for the semantic noise. Table 3 shows that NPCL achieves competitive performance compared with RoG for open-set noise.

Table 1: Test accuracy(%) of DenseNet on CIFAR10 and CIFAR100.

| Noise type | CIFAR10 | | | | CIFAR100 | | | |
|---|---|---|---|---|---|---|---|---|
| | NPCL | CL | Softmax | RoG | NPCL | CL | Softmax | RoG |
| uniform (20%) | **89.49** | 89.32 | 81.01 | 87.41 | 64.88 | **67.92** | 61.72 | 64.29 |
| uniform (40%) | 83.24 | **85.57** | 72.34 | 81.83 | 56.34 | **58.63** | 50.89 | 55.68 |
| uniform (60%) | 66.2 | 68.52 | 55.42 | **75.45** | 44.49 | **46.65** | 38.33 | 44.12 |

Table 2: Test accuracy(%) of DenseNet on CIFAR10 and CIFAR100 with semantic noise.

| Dataset | Label generator (noise rate) | NPCL | CL | Cross-entropy | D2L |
|---|---|---|---|---|---|
| CIFAR10 | DenseNet(32%) | 66.5 | **67.45** | 67.24 | 66.91 |
| | ResNet(38%) | 61.88 | **62.88** | 62.26 | 59.10 |
| | VGG(34%) | 68.37 | **69.61** | 68.77 | 57.97 |
| CIFAR100 | DenseNet(34%) | **57.59** | 55.14 | 50.72 | 5.00 |
| | ResNet(37%) | **54.49** | 53.20 | 50.68 | 23.71 |
| | VGG(37%) | **55.41** | 52.71 | 51.08 | 40.97 |

Table 3: Test accuracy(%) of DenseNet on CIFAR10 with open-set noise.

| Open-set Data | NPCL | Softmax | RoG |
|---|---|---|---|
| CIFAR100 | 82.85 | 79.01 | **83.37** |
| ImageNet | **87.95** | 86.88 | 87.05 |
| CIFAR100-ImageNet | 84.28 | 81.58 | **84.35** |

We further evaluate the performance of CL/NPCL on the Tiny-ImageNet dataset. We use the ResNet18 network as the test-bed. For GCE loss, we employ the default hyper-parameter $q = 0.7$ in all cases. All the methods are performed five runs with seeds $\{1, 2, 3, 4, 5\}$. The curve of mean test accuracy (shaded in std) are provided in Figure 2. We can see that NPCL and CL obtain higher test accuracy than generalized cross-entropy loss and stand cross-entropy loss on both cases. Note that CL does not have parameters, it is much convenient to use.

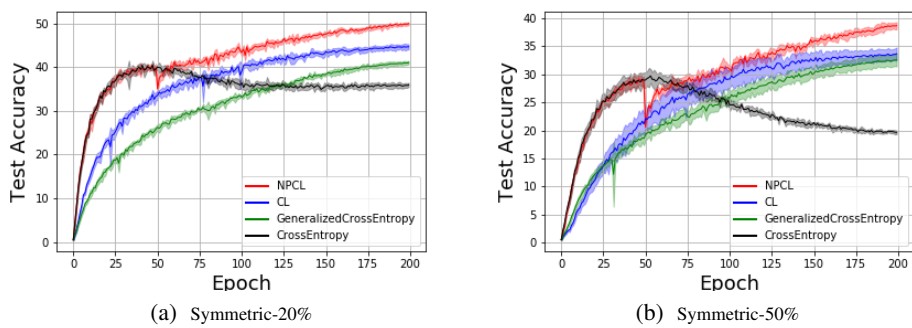

(a) Symmetric-20%      (b) Symmetric-50%

Figure 2:  Test accuracy (%) on Tiny-ImageNet dataset with symmetric noise

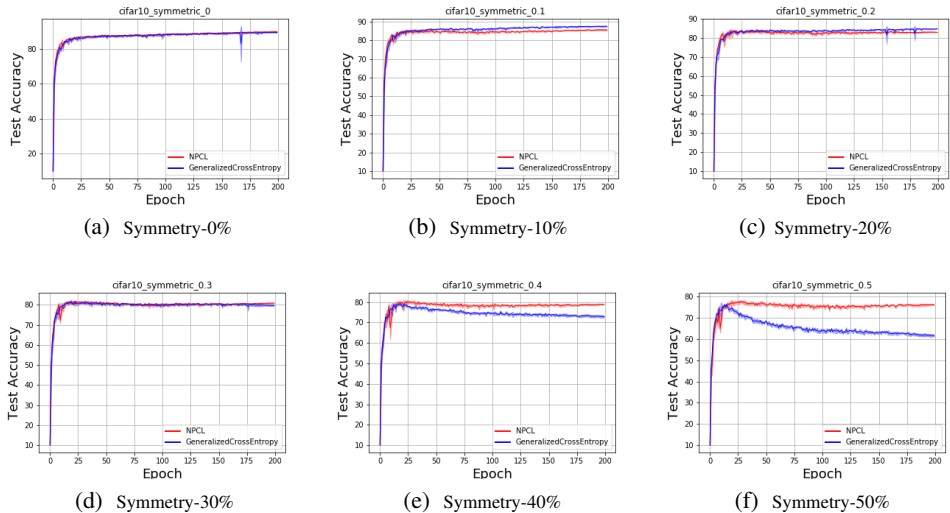

(a) Symmetry-0%      (b) Symmetry-10%      (c) Symmetry-20%

(d) Symmetry-30%      (e) Symmetry-40%      (f) Symmetry-50%

Figure 3: Test accuracy vs. number of epochs on CIFAR10 dataset.

## 4   CONCLUSION AND FURTHER WORK

In this work, we proposed a curriculum loss (CL) for robust learning. Theoretically, we analyzed the properties of CL and proved that it is tighter upper bound of the 0-1 loss compared with conventional summation based surrogate losses. We extended our CL to a more general form (NPCL) to handle large rate of label corruption. Empirically, experimental results on benchmark datasets show the robustness of the proposed loss. As a further work, we may improve our CL to handle imbalanced distribution by considering diversity for each class. Moreover, it is interesting to investigate the influence of different base loss functions in CL and NPCL.

## ACKNOWLEDGEMENT

We sincerely thank the reviewers for their insightful comments and suggestions. This paper was supported by Australian Research Council grants DP180100106 and DP200101328.

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

## A EXPLANATION OF THEOREM 1 FOR ROBUST LEARNING

**Theorem.** *(**Monotonic Relationship**) ((Hu et al., 2018) Let $p(x,y)$ and $q(x,y)$ be the training and test density,respectively. Define $r(x,y) = q(x,y)/p(x,y)$ and $r_i = r(x_i, y_i)$. Let $l(\widehat{y}, y) = \mathbf{1}\big(sign(\widehat{y}) \neq y\big)$ and $l(\widehat{y}, y) = \mathbf{1}\big(argmax_k(\widehat{y}_k) \neq y\big)$ be 0-1 loss for binary classification and multi-class classification, respectively. Let $f(\cdot)$ be convex with $f(1) = 0$. Define risk $\mathcal{R}(\theta)$, empirical risk $\widehat{\mathcal{R}}(\theta)$, adversarial risk $\mathcal{R}_{adv}(\theta)$ and empirical adversarial risk $\widehat{\mathcal{R}}_{adv}(\theta)$ as*

$$\mathcal{R}(\theta) = \mathbb{E}_{p(x,y)}\left[l(g_\theta(x), y)\right] \tag{22}$$

$$\widehat{\mathcal{R}}(\theta) = \frac{1}{n}\sum_{i=1}^{n} l(g_\theta(x_i), y_i) \tag{23}$$

$$\mathcal{R}_{adv}(\theta) = \sup_{r \in \mathcal{U}_f} \mathbb{E}_{p(x,y)}\left[r(x,y)l(g_\theta(x), y)\right] \tag{24}$$

$$\widehat{\mathcal{R}}_{adv}(\theta) = \sup_{\mathbf{r} \in \widehat{\mathcal{U}}_f} \frac{1}{n}\sum_{i=1}^{n} r_i l(g_\theta(x_i), y_i), \tag{25}$$

*where $\mathcal{U}_f = \big\{r(x,y)\,\big|\,\mathbb{E}_{p(x,y)}\left[f\left(r(x,y)\right)\right] \leq \delta, \mathbb{E}_{p(x,y)}\left[r(x,y)\right] = 1, r(x,y) \geq 0, \forall(x,y) \in \mathcal{X} \times \mathcal{Y}\big\}$ and $\widehat{\mathcal{U}}_f = \big\{\mathbf{r}\,\big|\,\frac{1}{n}\sum_{i=1}^{n} f(r_i) \leq \delta, \frac{1}{n}\sum_{i=1}^{n} r_i = 1, \mathbf{r} \geq 0\big\}$. Then we have that*

$$\textbf{If } \ \mathcal{R}_{adv}(\theta_1) < 1, \ \textbf{ then } \ \mathcal{R}(\theta_1) < \mathcal{R}(\theta_2) \iff \mathcal{R}_{adv}(\theta_1) < \mathcal{R}_{adv}(\theta_2). \tag{26}$$

$$\textbf{If } \ \mathcal{R}_{adv}(\theta_1) = 1, \ \textbf{ then } \ \mathcal{R}(\theta_1) \leq \mathcal{R}(\theta_2) \iff \mathcal{R}_{adv}(\theta_2) = 1. \tag{27}$$

*The same monotonic relationship holds between their empirical approximation: $\widehat{\mathcal{R}}(\theta)$ and $\widehat{\mathcal{R}}_{adv}$.*

Hu et al. (2018) show that minimizing (empirical) risk is equivalent to minimize the (empirical) adversarial risk (worst-case risk) for 0-1 loss. Thus, we can directly optimize the risk instead of the worst-case risk. Specifically, suppose we have an observable training distribution $p(x,y)$. The observable distribution $p(x,y)$ may be corrupted from an underlying clean distribution $q(x,y)$. We train a model based on the training distribution $p(x,y)$, and we want our model to perform well on the clean distribution $q(x,y)$. Since we do not know the clean distribution $q(x,y)$, we want our model to perform well for the worst-case estimate of the clean distribution, with the assumption that the $f$-divergence between the corrupted distribution $p$ and the clean distribution $q$ is bounded by $\delta$. Note that the underlying clean distribution is fixed but unknown, given the corrupted training distribution, the smallest $\delta$ that bounds the divergence between the corrupted distribution and clean distribution measures the intrinsic difficulty of the corruption, and it is also fixed and unknown. The corresponding worst-case distribution w.r.t the smallest $\delta$ is an estimate of the true clean distribution, and this worst-case risk upper bounds the risk of the true clean distribution. In addition, this bound is tighter than the other worst-case risks w.r.t larger $\delta$. Formally, the upper bound w.r.t the smallest $\delta$ is given as

$$G(\theta) := \sup_{q \in \widetilde{\mathcal{U}}_f} \mathbb{E}_{q(x,y)}\left[l(g_\theta(x), y)\right] \tag{28}$$

where $\widetilde{\mathcal{U}}_f$ is an equivalent constrainted set w.r.t $\mathcal{U}_f$ for $q(x,y)$. Then, we have

$$G(\theta) := \sup_{q \in \widetilde{\mathcal{U}}_f} \mathbb{E}_{q(x,y)}\left[l(g_\theta(x), y)\right] = \sup_{r \in \mathcal{U}_f} \mathbb{E}_{p(x,y)}\left[r(x,y)l(g_\theta(x), y)\right] \tag{29}$$

When $l(\cdot)$ is 0-1 loss, from Theorem 1, we know that minimize $G(\theta)$ is equivalent to minimize $\widetilde{G}(\theta)$. Thus, we can minimize $\widetilde{G}(\theta)$ instead of $G(\theta)$.

$$\widetilde{G}(\theta) := \mathbb{E}_{p(x,y)}\left[l(g_\theta(x), y)\right] \tag{30}$$

Minimize the Eq.(30) enables us to minimize the Eq.(28) without knowing the true divergence parameter $\delta$ beforehand. Usually, minimizing the upper bound can decrease the true risk under clean distribution. Particularly, when the clean distribution coincides with the worst-case estimate w.r.t the smallest $\delta$, minimizing the risk under the corrupted training distribution leads to the same minimizer as minimizing the risk under the clean distribution.

**Relationship between label corruption and general corruption**

Label corruption is a special case of general corruption. Label corruption restricts the corruption in the space $\mathcal{Y}$ instead of the space $\mathcal{X} \times \mathcal{Y}$. That is to say, the training distribution $p(x)$ is same as the clean distribution $q(x)$ over $\mathcal{X}$. Then, we have the robust risk for label corruption as

$$G_y(\theta) := \sup_{q \in \widetilde{\mathcal{U}}_f \cap H} \mathbb{E}_{q(x,y)}\left[l(g_\theta(x), y)\right] \tag{31}$$

where $H := \{q(x, y) \,|\, q(x) = p(x), \forall (x, y) \in \mathcal{X} \times \mathcal{Y}\}$. The supremum in $G_y(\theta)$ is taken over $\widetilde{\mathcal{U}}_f \cap H$, while the supremum in $G(\theta)$ is taken over $\widetilde{\mathcal{U}}_f$. Due to the additional constrain $q(x) = p(x), \forall (x, y) \in \mathcal{X} \times \mathcal{Y}$, we thus know that the robust risk $G_y(\theta)$ is bounded by $G(\theta)$, i.e., $G_y(\theta) \leq G(\theta)$. Moreover, it is more piratical and important to be robust for both label corruption and feature corruption.

## B  PROOF OF THEOREM 2

*Proof.* Because $\mathbf{1}(u < 0) \leq l(u)$, we have $\sum_{i=1}^n l(u_i) \geq \sum_{i=1}^n \mathbf{1}(u_i < 0)$. Then

$$Q(\mathbf{u}) = \min_{\mathbf{v} \in \{0,1\}^n} \max \left( \sum_{i=1}^n v_i l(u_i), n - \sum_{i=1}^n v_i + \sum_{i=1}^n \mathbf{1}(u_i < 0) \right) \tag{32}$$

$$\leq \max \left( \sum_{i=1}^n l(u_i), n - \sum_{i=1}^n 1 + \sum_{i=1}^n \mathbf{1}(u_i < 0) \right) \tag{33}$$

$$= \max \left( \sum_{i=1}^n l(u_i), \sum_{i=1}^n \mathbf{1}(u_i < 0) \right) \tag{34}$$

$$= \sum_{i=1}^n l(u_i) \tag{35}$$

Since loss $\widehat{J}(\mathbf{u}) = \sum_{i=1}^n l(u_i)$, we obtain $Q(\mathbf{u}) \leq \widehat{J}(\mathbf{u})$.

On the other hand, we have that

$$Q(\mathbf{u}) = \min_{\mathbf{v} \in \{0,1\}^n} \max \left( \sum_{i=1}^n v_i l(u_i), n - \sum_{i=1}^n v_i + \sum_{i=1}^n \mathbf{1}(u_i < 0) \right)$$

$$\geq \min_{\mathbf{v} \in \{0,1\}^n} n - \sum_{i=1}^n v_i + \sum_{i=1}^n \mathbf{1}(u_i < 0) \tag{36}$$

$$= \sum_{i=1}^n \mathbf{1}(u_i < 0) \tag{37}$$

Since $J(\mathbf{u}) = \sum_{i=1}^n \mathbf{1}(u_i < 0)$, we obtain $Q(\mathbf{u}) \geq J(\mathbf{u})$

$\square$

## C  PROOF OF COROLLARY 1

*Proof.* Since $n = mb$, similar to the proof of $Q(\mathbf{u}) \leq \widehat{J}(\mathbf{u})$, we have

$$\widehat{Q}(\mathbf{u}) = \sum_{j=1}^b \min_{\mathbf{v} \in \{0,1\}^m} \max \left( \sum_{i=1}^m v_{ij} l(u_{ij}), m - \sum_{i=1}^m v_{ij} + \sum_{i=1}^m \mathbf{1}(u_{ij} < 0) \right)$$

$$\leq \sum_{j=1}^b \max \left( \sum_{i=1}^m l(u_{ij}), m - \sum_{i=1}^m 1 + \sum_{i=1}^m \mathbf{1}(u_{ij} < 0) \right) \tag{38}$$

$$= \sum_{j=1}^b \max \left( \sum_{i=1}^m l(u_{ij}), \sum_{i=1}^m \mathbf{1}(u_{ij} < 0) \right) \tag{39}$$

$$= \sum_{j=1}^b \sum_{i=1}^m l(u_{ij}) = \widehat{J}(\mathbf{u}) \tag{40}$$

On the other hand, since the group (batch) separable sum structure, we have that

$$\widehat{Q}\left(\mathbf{u}\right) = \sum_{j=1}^{b} \min_{\mathbf{v} \in \{0,1\}^m} \max\left(\sum_{i=1}^{m} v_{ij} l(u_{ij}), m - \sum_{i=1}^{m} v_{ij} + \sum_{i=1}^{m} \mathbf{1}\left(u_{ij} < 0\right)\right)$$

$$= \min_{\mathbf{v} \in \{0,1\}^n} \sum_{j=1}^{b} \max\left(\sum_{i=1}^{m} v_{ij} l(u_{ij}), m - \sum_{i=1}^{m} v_{ij} + \sum_{i=1}^{m} \mathbf{1}\left(u_{ij} < 0\right)\right) \tag{41}$$

$$\geq \min_{\mathbf{v} \in \{0,1\}^n} \max\left(\sum_{j=1}^{b}\sum_{i=1}^{m} v_{ij} l(u_{ij}), n - \sum_{j=1}^{b}\sum_{i=1}^{m} v_{ij} + \sum_{j=1}^{b}\sum_{i=1}^{m} \mathbf{1}\left(u_{ij} < 0\right)\right) \tag{42}$$

$$= Q\left(\mathbf{u}\right) \geq J\left(\mathbf{u}\right) \tag{43}$$

$\square$

# D    PROOF OF PARTIAL OPTIMIZATION THEOREM (THEOREM 4)

*Proof.* For simplicity, let $l_i = l(u_i)$, $i \in \{1, ..., n\}$. Without loss of generality, assume $l_1 \leq l_2 \cdots \leq l_n$. Let $\mathbf{v}^*$ be the solution obtained by Algorithm 1. Assume there exits a $\mathbf{v}$ such that

$$\max\left(\sum_{i=1}^{n} v_i l_i, C - \sum_{i=1}^{n} v_i\right) < \max\left(\sum_{i=1}^{n} v_i^* l_i, C - \sum_{i=1}^{n} v_i^*\right). \tag{44}$$

Let $T = \sum_{i=1}^{n} v_i$ and $T^* = \sum_{i=1}^{n} v_i^*$.

**Case 1:** If $T = T^*$, then there exists an $v_k = 1$ and $v_k^* = 0$. From Algorithm 1, we know $k > T^*$ ($v_k^* = 0 \Rightarrow k > T^*$) and $l_k \geq l_j, j \in \{1, ..., T^*\}$. Then we know $\sum_{i=1}^{n} v_i^* l_i \leq \sum_{i=1}^{n} v_i l_i$. Thus, we can achieve that

$$\max\left(\sum_{i=1}^{n} v_i^* l_i, C - \sum_{i=1}^{n} v_i^*\right) = \max\left(\sum_{i=1}^{n} v_i^* l_i, C - \sum_{i=1}^{n} v_i\right) \tag{45}$$

$$\leq \max\left(\sum_{i=1}^{n} v_i l_i, C - \sum_{i=1}^{n} v_i\right). \tag{46}$$

This contradicts the assumption in Eq.(44)

**Case 2:** If $T > T^*$, then there exists an $v_k = 1$ and $v_k^* = 0$. Let $\mathrm{L}_{T^*} = \sum_{i=1}^{T^*} l_i$. Since $l_k \geq 0$, we have $\mathrm{L}_{T^*} + l_k \geq \mathrm{L}_{T^*}$. From Algorithm 1, we know that $\mathrm{L}_{T^*} + l_k > C - T^*$. Thus we obtain that

$$\max\left(\sum_{i=1}^{n} v_i l_i, C - \sum_{i=1}^{n} v_i\right) \geq \mathrm{L}_{T^*} + l_k \tag{47}$$

$$\geq \max\left(\mathrm{L}_{T^*}, C - T^*\right) \tag{48}$$

$$= \max\left(\sum_{i=1}^{n} v_i^* l_i, C - \sum_{i=1}^{n} v_i^*\right) \tag{49}$$

This contradicts the assumption in Eq.(44)

**Case 3:** If $T < T^*$, we obtain $C - T \geq C - T^* + 1$. Then we can achieve that

$$\max\left(\sum_{i=1}^{n} v_i^* l_i, C - \sum_{i=1}^{n} v_i^*\right) = \max\left(\mathrm{L}_{T^*}, C - T^*\right) \tag{50}$$

$$\leq C + 1 - T^* \tag{51}$$

$$\leq C - T \tag{52}$$

$$= C - \sum_{i=1}^{n} v_i \tag{53}$$

$$\leq \max\left(\sum_{i=1}^{n} v_i l_i, C - \sum_{i=1}^{n} v_i\right). \tag{54}$$

This contradicts the assumption in Eq.(44).

Finally, we conclude that $\mathbf{v}^*$ obtained by Algorithm 1 is the minimum of the optimization problem given in (13). $\qquad\square$

## E  PROOF OF PROPOSITION 1

*Proof.* Note that $T^* = \sum_{i=1}^{n} v_i^*$, from the condition of $v_i^* = 1$ in Algorithm 1, we know that $L_{T^*} \leq C+1-T^*$. From the condition of $v_k^* = 0$ in Algorithm 1, we know that $L_{T^*+1} > C-T^*$. Because $l(u_i) \geq \mathbf{1}(u_i < 0) \geq 0$ for $i \in \{1, ..., n\}$, we have $L_{T^*+1} = L_{T^*} + l(u_{T^*+1}) \geq L_{T^*}$. Thus, we obtain $L_{T^*+1} > \max(L_{T^*}, C - T^*)$. By substitute the optimum $\mathbf{v}^*$ into the optimization function, we obtain that

$$\min_{\mathbf{v} \in \{0,1\}^n} \max \left( \sum_{i=1}^{n} v_i l(u_i), C - \sum_{i=1}^{n} v_i \right) \tag{55}$$

$$= \max \left( \sum_{i=1}^{n} v_i^* l(u_i), C - \sum_{i=1}^{n} v_i^* \right) \tag{56}$$

$$= \max(L_{T^*}, C - T^*) \tag{57}$$

$\qquad\square$

## F  PROOF OF THEOREM 3

*Proof.* We first prove that objective (11) is tighter than the loss objective $\widehat{J}(\mathbf{u})$ in Eq.(8). After this, we prove that objective (11) is an upper bound of the 0/1 loss defined in equation (7).

For simplicity, let $l_i = l(u_i)$, we obtain that

$$E(\mathbf{u}) = \min_{\mathbf{v} \in \{0,1\}^n} \max(\sum_{i=1}^{n} v_i l(u_i), n - \sum_{i=1}^{n} v_i) \tag{58}$$

$$\leq \max(\sum_{i=1}^{n} l(u_i), (n - \sum_{i=1}^{n} 1)) \tag{59}$$

$$= \sum_{i=1}^{n} l(u_i). \tag{60}$$

Note that $\widehat{J}(\mathbf{u}) = \sum_{i=1}^{n} l(u_i)$, thus, we have $E(\mathbf{u}) \leq \widehat{J}(\mathbf{u})$.

Without loss of generality, assume $l_1 \leq l_2 \cdots \leq l_n$. Let $L_i = \sum_{j=1}^{i} l_j$, $T = \sum_{i=1}^{n} v_i^*$, where $\mathbf{v}^* = [v_1^*, v_2^* \cdots v_n^*]^T$ is the optimum of $v$ for fixed $\mathbf{u}$. Let $k = \sum_{i=1}^{n} \mathbf{1}(u_i \geq 0)$. Then we achieve that the 0/1 loss $J(\mathbf{u})$ is as follows:

$$J(\mathbf{u}) = \sum_{i=1}^{n} \mathbf{1}(u_i < 0) = n - k. \tag{61}$$

From Algorithm 1 with $C = n$, we achieve that $L_T \leq n - T + 1$ and $L_{T+1} > n - T$.

**Case 1:** If $k \geq T$, we can achieve that

$$2E(\mathbf{u}) - J(\mathbf{u}) = 2\max(L_T, n - T) - (n - k) \tag{62}$$

$$\geq 2(n - T) - (n - k) \tag{63}$$

$$= n + k - 2T \geq 0.$$

**Case 2:** If $k < T, n - T \geq L_T$, we can obtain that

$$2E\left(\mathbf{u}\right) - J(\mathbf{u}) = 2(n - T) - (n - k) = n + k - 2T. \tag{64}$$

Since $k < T$, if follows that

$$L_T = L_k + \sum_{j=k+1}^{T} l_j \geq L_k + \sum_{j=k+1}^{T} 1 \tag{65}$$

$$= L_k + T - k \tag{66}$$

$$\geq T - k. \tag{67}$$

Together with $n - T \geq L_T$ , we can obtain that

$$n - T \geq L_T \geq T - k \Rightarrow n + k - 2T \geq 0. \tag{68}$$

Thus, we can achieve that

$$2E\left(\mathbf{u}\right) - J(\mathbf{u}) = n + k - 2T \geq 0. \tag{69}$$

**Case 3:** If $k < T, n - T < L_T$, we can obtain that

$$2E\left(\mathbf{u}\right) - J(\mathbf{u}) = 2\max(L_T, n - T) - (n - k) \tag{70}$$

$$= 2L_T - (n - k) \tag{71}$$

$$> (n - T) + L_T - n + k. \tag{72}$$

From (67), we have $L_T \geq T - k$. Together with (72), it follows that

$$2E\left(\mathbf{u}\right) - J(\mathbf{u}) > (n - T) + (T - k) - n + k \geq 0. \tag{73}$$

Finally, we can achieve that $J(\mathbf{u}) \leq 2E\left(\mathbf{u}\right) \leq 2\widehat{J}\left(\mathbf{u}\right)$ . $\qquad\square$

## G  PROOF OF COROLLARY 2

*Proof.* Since $n = mb$, similar to the proof of $\widehat{Q}\left(\mathbf{u}\right) \leq \widehat{J}\left(\mathbf{u}\right)$, we have

$$\widehat{E}\left(\mathbf{u}\right) = \sum_{j=1}^{b} \min_{\mathbf{v} \in \{0,1\}^m} \max\left(\sum_{i=1}^{m} v_{ij} l(u_{ij}), m - \sum_{i=1}^{m} v_{ij}\right)$$

$$\leq \sum_{j=1}^{b} \max\left(\sum_{i=1}^{m} l(u_{ij}), m - \sum_{i=1}^{m} 1\right) \tag{74}$$

$$= \sum_{j=1}^{b} \max\left(\sum_{i=1}^{m} l(u_{ij}), 0\right) \tag{75}$$

$$= \sum_{j=1}^{b} \sum_{i=1}^{m} l(u_{ij}) = \widehat{J}\left(\mathbf{u}\right) \tag{76}$$

On the other hand, since the group (batch) separable sum structure, we have that

$$\widehat{E}\left(\mathbf{u}\right) = \sum_{j=1}^{b} \min_{\mathbf{v} \in \{0,1\}^m} \max\left(\sum_{i=1}^{m} v_{ij} l(u_{ij}), m - \sum_{i=1}^{m} v_{ij}\right)$$

$$= \min_{\mathbf{v} \in \{0,1\}^n} \sum_{j=1}^{b} \max\left(\sum_{i=1}^{m} v_{ij} l(u_{ij}), m - \sum_{i=1}^{m} v_{ij}\right) \tag{77}$$

$$\geq \min_{\mathbf{v} \in \{0,1\}^n} \max\left(\sum_{j=1}^{b} \sum_{i=1}^{m} v_{ij} l(u_{ij}), n - \sum_{j=1}^{b} \sum_{i=1}^{m} v_{ij}\right) \tag{78}$$

$$= E\left(\mathbf{u}\right) \tag{79}$$

Together with Theorem 3, we obtain that $J(\mathbf{u}) \leq 2E\left(\mathbf{u}\right) \leq 2\widehat{E}\left(\mathbf{u}\right) \leq 2\widehat{J}\left(\mathbf{u}\right)$

$\qquad\square$

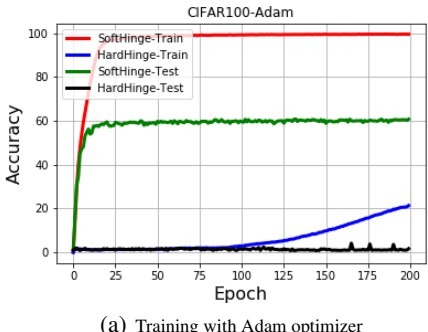 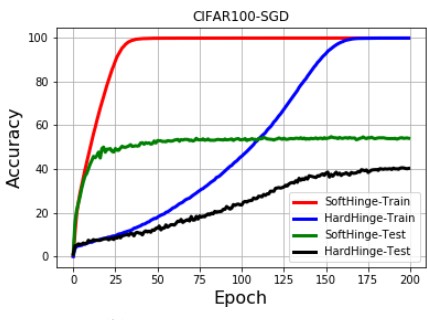

(a) Training with Adam optimizer  (b) Training with SGD optimizer

Figure 4:  Training/Test accuracy for soft and hard hinge loss with different optimizer on CIFAR100

## H  MULTI-CLASS EXTENSION

For multi-class classification, denote the groudtruth label as $y \in \{1, ..., K\}$. Denote the classification prediction (the last layer output of networks before loss function) as $t_i, i \in \{1, ..., K\}$. Then, the classification margin for multi-class classification can be defined as follows

$$u = t_y - \max_{i \neq y} t_i. \tag{80}$$

We can see that $\mathbf{1}(u < 0) = \mathbf{1}(t_y - \max_{i \neq y} t_i < 0)$ is indeed the 0-1 loss for multi-class classification.

With the classification margin $u$, we can compute the base loss $l(u) \geq \mathbf{1}(u < 0)$. In this work, we employ the hinge loss. As we need the upper bound of 0-1 loss, the multi-class hard hinge loss function Moore & DeNero (2011) can be defined as

$$H(\mathbf{t}, y) = \max(1 - u, 0) = \max(1 - t_y + \max_{i \neq y} t_i, 0). \tag{81}$$

The multi-class hard hinge loss in Eq.(81) is not easy to optimize for deep networks. We propose a novel soft multi-class hinge loss function as follows:

$$S(\mathbf{t}, y) = \begin{cases} \max(1 - t_y + \max_{i \neq y} t_i, 0) & , t_y - \max_{i \neq y} t_i \geq 0 \\ \max(1 - t_y + \text{LogSumExp}(\mathbf{t}), 0) & , t_y - \max_{i \neq y} t_i < 0. \end{cases} \tag{82}$$

The soft hinge loss employs the LogSumExp function to approximate the max function when the classification margin is less than zero, i.e., misclassification case. Intuitively, when the sample is misclassified, it is far away from being correctly separate by a positive margin (e.g. margin $u \geq 1$). In this situation, a smooth loss function can help speed up gradient update. Because $\text{LogSumExp}(\mathbf{t}) > \max_{i \in \{1, \cdots K\}} t_i$ we know that the soft hinge loss is an upper bound of the hard hinge loss, i.e., $S(\mathbf{t}, y) \geq H(\mathbf{t}, y)$ . Moreover, we can obtain a new weighted loss $F(\mathbf{t}, y; \beta) = \beta S(\mathbf{t}, y) + (1 - \beta) H(\mathbf{t}, y), \beta \in [0, 1]$ that is also an upper bound of 0-1 loss.

## I  EVALUATION OF EFFICIENCY OF THE PROPOSED SOFT-HINGE LOSS

We compare our soft multi-class hinge loss with hard multi-class hinge loss Moore & DeNero (2011) on CIFAR100 dataset training with Adam and SGD optimizer, respectively. We keep both the network architecture and hyperparameters same. We employ the default learning rate and momentums of Adam optimizer in PyTorch toolbox, i.e. $lr = 10^{-3}, \beta_1 = 0.9, \beta_2 = 0.999$. For SGD optimizer, the learning rate $(lr)$ and momentum $(\rho)$ are set to $lr = 10^{-2}$ and $\rho = 0.9$ respectively.

The pictures of training/test accuracy v.s number of epochs are presented in Figure 4. We can observe that both the training accuracy and the test accuracy of our soft hinge loss increase greatly fast as the number of epochs increase. In contrast, the training and test accuracy of hard hinge loss grow very slowly. The training accuracy of soft hinge loss can arrive $100\%$ trained with both optimizers. Both

training and test accuracy of soft hinge loss are consistently better than hard hinge loss. In addition, training accuracy of hard hinge loss can also reach $100\%$ when SGD optimizer is used. However, its test accuracy is lowever than that of soft hinge loss.

## J    IMPACT OF MISSPECIFIED ESTIMATION OF NOISE RATE $\epsilon$

We empirically analyze the impact of misspecified prior for the noise rate $\epsilon$. The average test accuracy over last ten epochs on MNIST for different priors are reported in Table 4. We can observe that NPCL is robust to misspecified prior for small noise cases (Symmetry-20%). Moreover, it becomes a bit more sensitive on large noise case (Symmetry-50%) and on the pair flipping case (Pair-35%).

Table 4: Average test accuracy of NPCL with different $\epsilon$ on MNIST over last ten epochs

| Flipping Rate | $0.5\epsilon$ | $0.75\epsilon$ | $\epsilon$ | $1.25\epsilon$ | $1.5\epsilon$ |
|---|---|---|---|---|---|
| Symmetry-20% | $96.31\% \pm 0.17\%$ | $97.72\% \pm 0.09\%$ | $99.41\% \pm 0.01\%$ | $\mathbf{99.55\% \pm 0.02\%}$ | $99.10\% \pm 0.04\%$ |
| Symmetry-50% | $78.67\% \pm 0.36\%$ | $87.36\% \pm 0.29\%$ | $\mathbf{98.53\% \pm 0.02\%}$ | $97.92\% \pm 0.06\%$ | $67.61\% \pm 0.06\%$ |
| Pair-35% | $80.59\% \pm 0.40\%$ | $87.86\% \pm 0.48\%$ | $97.90\% \pm 0.04\%$ | $\mathbf{99.33\% \pm 0.02\%}$ | $86.66\% \pm 0.08\%$ |

Table 5: Average test accuracy on MNIST over the last ten epochs.

| Flipping-Rate | Standard | MentorNet | Co-teaching | Co-teaching+ | GCE | NPCL |
|---|---|---|---|---|---|---|
| Symmetry-20% | $93.78\% \pm 0.04\%$ | $96.68\% \pm 0.05\%$ | $97.14\% \pm 0.03\%$ | $\mathbf{99.41\% \pm 0.01\%}$ | $99.40 \pm 0.01\%$ | $\mathbf{99.41\% \pm 0.01\%}$ |
| Symmetry-50% | $65.81\% \pm 0.14\%$ | $90.53\% \pm 0.07\%$ | $91.35\% \pm 0.09\%$ | $97.79\% \pm 0.03\%$ | $92.48 \pm 0.13\%$ | $\mathbf{98.53\% \pm 0.02\%}$ |
| Pair-35% | $70.50\% \pm 0.16\%$ | $89.62\% \pm 0.15\%$ | $90.96\% \pm 0.18\%$ | $93.81\% \pm 0.20\%$ | $72.26 \pm 0.06\%$ | $\mathbf{97.90\% \pm 0.04\%}$ |

Table 6: Average test accuracy on CIFAR10 over the last ten epochs.

| Flipping-Rate | Standard | MentorNet | Co-teaching | Co-teaching+ | GCE | NPCL |
|---|---|---|---|---|---|---|
| Symmetry-20% | $76.62\% \pm 0.07\%$ | $81.20\% \pm 0.09\%$ | $82.13\% \pm 0.08\%$ | $80.64\% \pm 0.15\%$ | $\mathbf{84.68\% \pm 0.05\%}$ | $84.30\% \pm 0.07\%$ |
| Symmetry-50% | $49.92\% \pm 0.09\%$ | $72.09\% \pm 0.06\%$ | $74.28\% \pm 0.11\%$ | $58.43\% \pm 0.30\%$ | $61.80\% \pm 0.11\%$ | $\mathbf{77.66\% \pm 0.09\%}$ |
| Pair-35% | $62.26\% \pm 0.09\%$ | $71.52\% \pm 0.06\%$ | $\mathbf{77.77\% \pm 0.14\%}$ | $62.72\% \pm 0.23\%$ | $60.86\% \pm 0.05\%$ | $76.52\% \pm 0.11\%$ |

## K    RELATED LITERATURE

**Curriculum Learning:** Curriculum learning is a general learning methodology that achieves success in many area. The very beginning work of curriculum learning (Bengio et al., 2009) trains a model gradually with samples ordered in a meaningful sequence, which has improved performance on many problems. Since the curriculum in (Bengio et al., 2009) is predetermined by prior knowledge and remained fixed later, which ignores the feedback of learners, Kumar et al. (Kumar et al., 2010) further propose Self-paced learning that selects samples by alternative minimization of an augmented objective. Jiang et al. (Jiang et al., 2014) propose a self-paced learning method to select samples with diversity. After that, Jiang et al. (Jiang et al., 2015) propose a self-paced curriculum strategy that takes different priors into consideration. Although these methods achieve success, the relation between the augmented objective of self-paced learning and the original objective (e.g. cross entropy loss for classification) is not clear. In addition, as stated in (Jiang et al., 2018), the alternative update in self-paced learning is not efficient for training deep networks.

**Learning with Noisy Labels:** The most related works are the sample selection based methods for robust learning. This kind of works are inspired by curriculum learning (Bengio et al., 2009). Among them, Jiang et al. (Jiang et al., 2018) propose to learn the curriculum from data by a mentor net. They use the mentor net to select samples for training with noisy labels. Co-teaching (Han et al., 2018b) employs two networks to select samples to train each other and achieve good generalization performance against large rate of label corruption. Co-teaching+ (Yu et al., 2019) extends Co-teaching by selecting samples with disagreement of prediction of two networks. Compared with Co-teaching/Co-teaching+, our CL is a simple plugin for a single network. Thus both space and time complexity of CL are half of Co-teaching's. Recently, Zhang & Sabuncu (2018) propose a generalized Cross-entropy loss for robust learning.

**Construction of tighter bounds of 0-1 loss:** Along the line of construction of tighter bounds of the 0-1 loss, many methods have been proposed. To name a few, Masnadi-Shirazi et al. (Masnadi-Shirazi & Vasconcelos, 2009) propose Savage loss, which is a non-convex upper bound of the 0-1

Table 7: Average test accuracy on CIFAR100 over the last ten epochs.

| Flipping-Rate | Standard | MentorNet | Co-teaching | Co-teaching+ | GCE | NPCL |
|---|---|---|---|---|---|---|
| Symmetry-20% | 47.05% ± 0.11% | 51.58% ± 0.15% | 53.89% ± 0.09% | **56.15% ± 0.09%** | 51.86% ± 0.09% | 55.30% ± 0.09% |
| Symmetry-50% | 25.47% ± 0.07% | 39.65% ± 0.10% | 41.08% ± 0.07% | 37.88% ± 0.06% | 37.60% ± 0.08% | **42.56% ± 0.06%** |
| Pair-35% | 39.91% ± 0.11% | 40.42% ± 0.07% | 43.36% ± 0.08% | 40.88% ± 0.16% | 36.64% ± 0.07% | **44.43% ± 0.15%** |

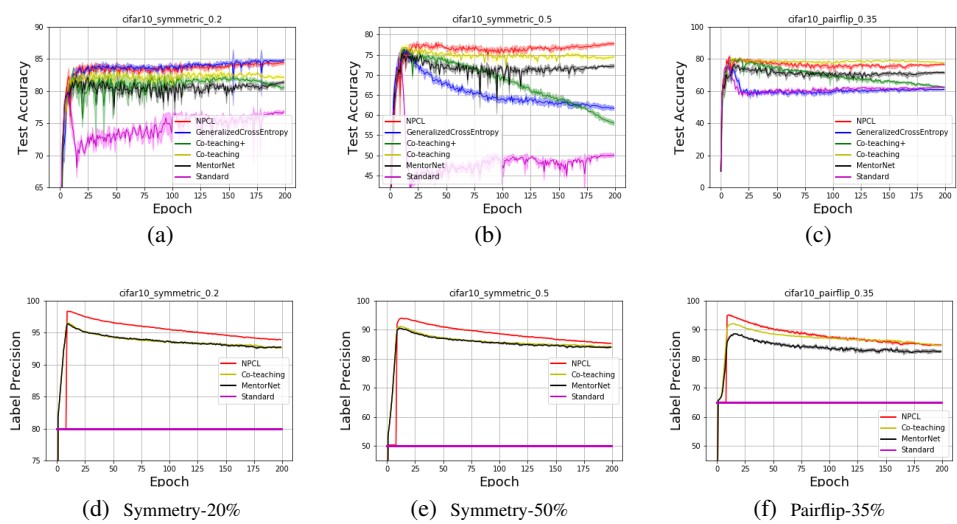

Figure 5: Test accuracy and label precision vs. number of epochs on CIFAR10 dataset.

loss function. Bartlett et al. (Bartlett et al., 2006) analyze the properties of the truncated loss for conventional convex loss. Wu et al. (Wu & Liu, 2007) study the truncated hinge loss for SVM. Although the results are fruitful, these works are mainly focus on loss function at individual data point, they do not have sample selection property. In contrast, our curriculum loss can automatically select samples for training. Moreover, it can be constructed in a tighter way than these individual losses by employing them as the base loss function.

## L ARCHITECTURE OF NEURAL NETWORKS

| CNN on MNIST | CNN on CIFAR-10 | CNN on CIFAR-100 |
|---|---|---|
| 28×28 Gray Image | 32×32 RGB Image | 32×32 RGB Image |
| | 3×3 conv, 128 LReLU | |
| | 3×3 conv, 128 LReLU | |
| | 3×3 conv, 128 LReLU | |
| | 2×2 max-pool, stride 2 | |
| | dropout, p = 0.25 | |
| | 3×3 conv, 256 LReLU | |
| | 3×3 conv, 256 LReLU | |
| | 3×3 conv, 256 LReLU | |
| | 2×2 max-pool, stride 2 | |
| | dropout, p = 0.25 | |
| | 3×3 conv, 512 LReLU | |
| | 3×3 conv, 256 LReLU | |
| | 3×3 conv, 128 LReLU | |
| | avg-pool | |
| dense 128→10 | dense 128→10 | dense 128→100 |

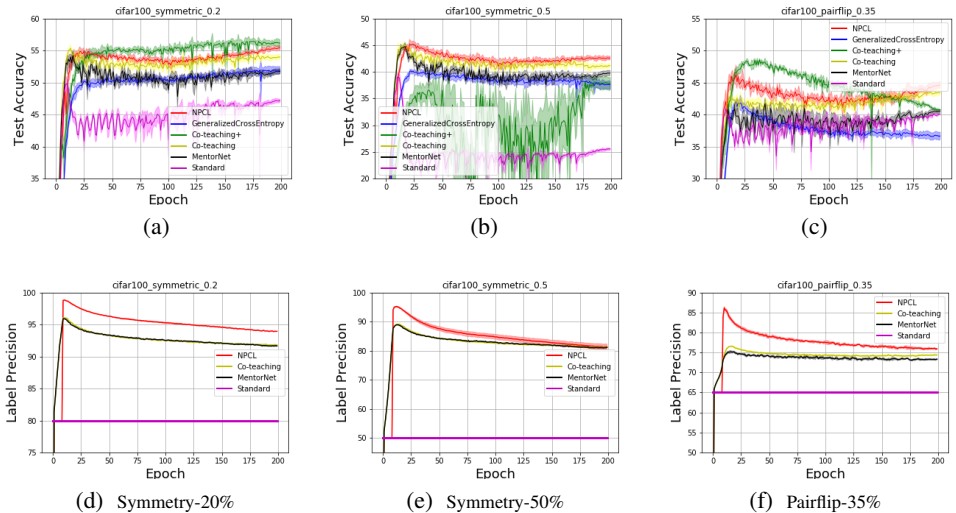

Figure 6: Test accuracy and label precision vs. number of epochs on CIFAR100 dataset.

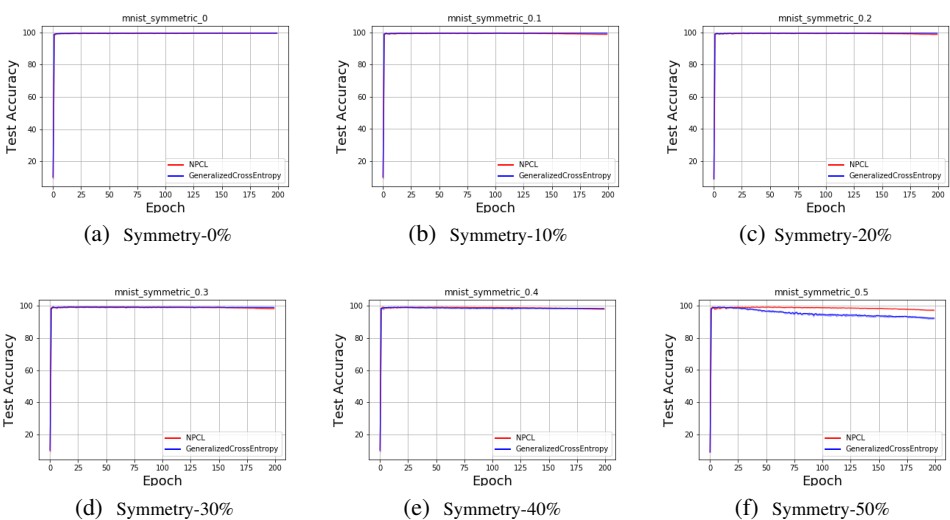

Figure 7: Test accuracy vs. number of epochs on MNIST dataset.

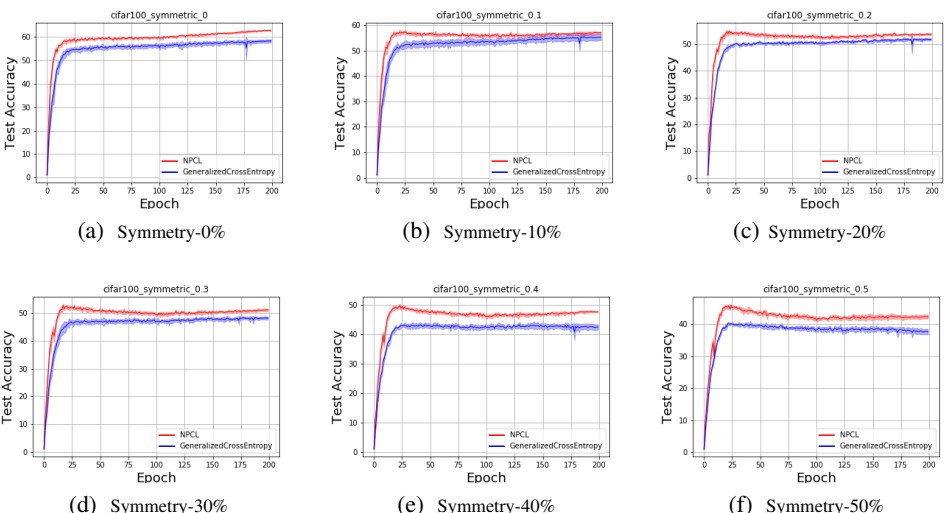

Figure 8: Test accuracy vs. number of epochs on CIFAR100 dataset.

