# OpenReview forum: "Curriculum Loss: Robust Learning and Generalization  against Label Corruption"
_ICLR.cc/2020/Conference — Accept (Poster)_

### Official Review · AnonReviewer3 · 2019-10-21
**Official Blind Review #3**

**Rating:** 6

**Review:**

Summary: This paper proposes a new loss function: curriculum loss, which is a meta-loss function that we can still specify an existing surrogate loss to use this loss function. This meta-loss function guarantees to be tighter than using a traditional pointwise-sum loss function as used in the empirical risk minimization framework. Intuitively, the proposed CL loss embed the sample selection process in the objective function. The authors suggest that it is robust against label corruption because it is tighter and provided promising experimental results.

========================================================
Clarity:
The paper is well-written and easy to follow.

========================================================
Significance:
The proposed paradigm is interesting and I am convinced that it can be useful under label noise. The experiments look promising. Future work about the analysis of NPCL/CL is also interesting to consider (e.g., which surrogate loss to use, rigorous theoretical guarantee, etc.). I think the proposed method is impactful.

========================================================
Comments:
The proposed method is interesting and can give a tighter bound for any surrogate loss by using this method (CL). Moreover, the author suggested a simple extension of CL for label corruption (NPCL) and the performance is impressive. I would like to vote accept for this paper but the following point highly concerns me and I am not sure about the correctness (see the concern below). It is about the motivation not the proposed method.

Concerns about motivation:

I disagree with the original motivation of this paper. The authors used the result of Hu et al. 2018 to motivate the use of CL. To my knowledge, the main point raised by Hu et al. is as follows:

In classification, minimizing the adversarial risk yields the same solution as using the standard empirical risk. This suggests that minimizing the adversarial risk may not enhance the robustness of a classifier. Yet, it may still be useful when we consider regression (other settings but not classification). As a result, in classification, we should try other methods to make a robust classifier. Then, Hu et al. considered to utilize some kind of structural assumption to make a robust classifier. From their title: "Does Distributionally Robust Supervised Learning Give Robust Classifiers?", I think they suggested "No" as an answer and the discussion about 0-1 loss in the curriculum loss paper will be contradicted to them from the motivation perspective.

Furthermore, regarding the adversarial risk, it is not focusing on the label noise but rather the noise of the feature-label pair, i.e, perturb (x,y) adversarially within an f-divergence ball. However, in my opinion, if we randomly flip the label of the data regardless of x (as the authors and existing work did in experiments when considering label corruption: symmetric, partial, etc.), we cannot be confident to state that the f-divergence between test distribution and corrupted training distribution is small under label noise.

Another point to motivate the use of 0-1 loss that the author mentioned is when we have outliers (Masnadi-Shirazi & Vasconcelos, 2009). This makes sense and this is a famous argument to discourage the use of too steep loss functions, e.g., exponential loss. I think this motivation is fine but it is not directly related to label corruption because we do not add out-of-distribution data but rather the label noise. Furthermore, the authors did not inject any outliers in the experiments in my understanding. I think this is totally no problem because we are focusing on label noise here, but this makes the motivation about outliers less important when we are talking about label noise.

I think the most important direction both in theory and experiments about the robustness to label noise of the 0-1 loss is that 0-1 loss satisfies a "symmetric property", i.e., \ell(z)+\ell(-z) = Constant for a margin-based loss function in binary classification. Under symmetric label noise, "the minimizer of the expected symmetric noise risk (a risk that the label is corrupted by coin flipping noise) is identical to the minimizer of the clean risk (normal risk)". Although it is not empirically but the expected version, it gives a good insight about the advantage of directly minimizing 0-1 loss under label noise. This is first pointed out by

[1] Manwani et al.: Noise tolerance under risk minimization, IEEE Transactions on Cybernetics 43 (2013)
[2] Ghosh et al.: Making risk minimization tolerant to label noise Neurocomputing 160 (2015): 93-107.

([1] focused on the 0-1 loss while [2] extended it to symmetric losses.)

Then, it was extended to the multiclass loss by the following paper:

[3] Ghosh et al.: Robust loss functions under label noise for deep neural networks. AAAI2017.

The advantage of symmetric losses is also discussed in this paper that the authors already cited in the symmetric noise experiment section.

[4] van Rooyen et al.: Learning with symmetric label noise: The importance of being unhinged, NeurIPS2015

The advantage of the symmetric condition and 0-1 loss is also discussed in a more general noise scenario and more evaluation metrics:

[5] van Rooyen et. al: An average classification algorithm. arXiv:1506.01520, 2015
[6] Charoenphakdee et al.: On symmetric losses for learning from corrupted labels, ICML2019

And the following paper that was also cited in the submitted work and compared:

[7] Zhang and Sabuncu: Generalized cross-entropy loss for training deep neural networks with noisy labels, NeurIPS2018

is also inspired by the robustness of the symmetric losses (including 0-1 loss). They argued that although the symmetric loss (MAE) for multiclass proposed by Ghosh AAAI2017 is robust, it is hard to train for challenging datasets, and they try to relevate this condition while making it easier to train.  This paper outperformed [7] and I think it is clearer and better to build a story along this line.

In short, here is the key message why I think the current motivation does not feel right. When we have noisy labeled data, instead of motivating the use of 0-1 loss by suggesting that

"If we have clean labeled data, minimizing the "adversarial" ERM risk using "clean" labeled data yields the same minimizer as minimizing the "standard" ERM risk using "clean" labeled data",

I believe the story to motivate the robustness of 0-1 loss under label noise should be

"If we have noisy labeled data, minimizing the "standard" or "modified" risk using "noisy" labeled data yields the same minimizer as minimizing the "standard" ERM risk using "clean" label data"

The latter statement corresponds to the literature I suggested.

Apart from the motivation raised by the authors, as we can see from this curriculum loss paper, NPCL nicely outperformed generalized cross entropy loss in [7], which is impressive.

========================================================
Decision.
I strongly feel that motivating the noise robustness of 0-1 loss by discussing about the adversarial risk (Hu et al.) is misleading. Nevertheless, I feel the proposed method itself makes a lot of sense and I am impressed by the results. If the author can convince me that using the current motivation of the paper is suitable, I am happy to improve the score. Another way is to agree to modify the motivation part. Given the experiments were done, it is not to difficult to change the motivation of the paper. At this point, I have decided to give a weak reject.

========================================================
Questions:
1. Is it straightforward to combine NPCL with Co-teaching/Mentornet/Co-teaching+?
2. Does the traditional theory about classification-calibration (Zhang, 2004, Bartlett+, 2006) can guarantee the Bayes-optimal solution if we use NPCL?

========================================================
Minor comments:
1. Page 9: Both our NPCL and Generalized Cross Entropy(GCE) << space missing between Entropy and (

Update: I have read the rebuttal. Although I am still not fully convinced with the motivation of the paper and still doubting whether NPCL works well because of the given motivation, I still believe that the proposed NPCL should give a new perspective to deal with noisy labels. I like the idea of the paper. Thus, I change the score to Weak Accept.


**Experience Assessment:**

I have published one or two papers in this area.

**Review Assessment: Checking Correctness Of Derivations And Theory:**

I assessed the sensibility of the derivations and theory.

**Review Assessment: Checking Correctness Of Experiments:**

I carefully checked the experiments.

**Review Assessment: Thoroughness In Paper Reading:**

I read the paper thoroughly.

---

> ### Author Response · Authors · 2019-11-15
> **Clarify the motivation.**
>
>
> Thanks for your comments.  We acknowledge your concern. Here, we want to clarify some misunderstandings about our motivation.
>
> Our motivation is not "If we have clean labeled data, minimizing the "adversarial" ERM risk using "clean" labeled data yields the same minimizer as minimizing the "standard" ERM risk using "clean" labeled data."  The reviewer’s concern is training with the clean distribution. However,  our motivation is to train on a corrupted training distribution p(x,y),  not a clean distribution, and we want our model to perform well on the worst-case clean distribution (in the f-divergence ball).
>
> Compared with Hu et al., our motivation is not training with  "adversarial" ERM to improve training with classification risk under the clean distribution.  Our key finding is that minimizing the classification risk under a corrupted distribution can minimize the classification risk of the worst-case clean distribution(in the f-divergence ball). There is no contradiction to Hu et al. Note that the worst-case classification risk is an upper bound of the classification risk of the true clean distribution, minimizing the worst-case classification risk can usually decrease the true classification risk.
>
> Specifically, suppose we have an observable training distribution p(x,y). The observable distribution p(x,y) may be corrupted from an underlying clean distribution q(x,y). We train a model based on the training distribution p(x,y), but we want our model to perform well on the clean distribution q(x,y). Since we do not know the clean distribution q(x,y), we want our model to perform well even for the worst-case clean distribution q, with the assumption that the f-divergence between the corrupted distribution p and the clean distribution q is bounded by delta.   Because of Theorem 1, we do not need to optimize the worst-case risk directly; we can optimize the classification risk (on the corrupted training distribution) instead.
>
> We provide a more detailed explanation in Appendix A and update the paper to make the motivation clear.
>
>
> Thanks for the suggestion of another line of analysis of the robustness of 0-1 loss.
>
> Actually, the "symmetric property" of robust loss is derived under additional assumptions of noise type. For example, In Ghosh AAAI2017, they make assumptions of uniform noise,  simple non-uniform noise, and class conditional noise.
>
> If we assume the noise is uniform, minimizing the (empirical) risk of 0-1 loss using noisy data leads to a same minimizer as minimizing the (empirical) risk for the clean distribution. (Ghosh AAAI2017)
>
> If we do not assume the noise type, minimizing the risk of 0-1 loss using noisy data leads to a same minimizer as minimizing the risk of the worst-case clean distribution (in the f-divergence ball), which is the case of this work.
>
> The robustness of 0-1 is interesting; we will further analyze this robustness in further work.

---

> > ### Comment · AnonReviewer3 · 2019-11-20
> > **On the motivation**
> >
> > Thank you for the clarification! I would like to clarify that in my previous review about motivation, I  did not misunderstand the motivation but I want to emphasize that the message of Theorem 1 basically says that
> >
> > the minimizer of the clean distribution is identical to that of the worst-case distribution around that clean distribution (which refers to the f-divergence ball).
> >
> > And I acknowledged that the authors wanted to interpret as
> >
> > the minimizer of the corrupted distribution is identical to that of the worst-case "clean" distribution around that corrupted distribution.
> >
> > My concern is that if what the author suggested is true, although it is free from noise assumption, it sounds like the minimizer of the corrupted risk w.r.t. 0-1 loss is identical to the worst-case clean distribution with arbitrary delta (which determines the size of the f-divergence ball). This sounds highly pessimistic if I did not misunderstand this part. Could you please clarify this part?
> >
> > Regarding the key finding of authors in the rebuttal:
> > "Our key finding is that minimizing the classification risk under a corrupted distribution can minimize the classification risk of the worst-case clean distribution (in the f-divergence ball)."
> >
> > I think this is the claim from the experimental results. I am not sure if the success of the proposed method is really because of that finding the author suggested, but the loss itself has some mechanics that make it robust to noise, e.g., adaptive sample selection.
> >
> > Although I am still not fully convinced with the motivation of the paper and still doubting whether NPCL works well because of the given motivation, I still believe that the proposed NPCL should improve the performance and also give a new perspective to deal with noisy labels. I like the idea of the paper. Thus, I increased my score.

---

### Official Review · AnonReviewer4 · 2019-10-23
**Official Blind Review #4**

**Rating:** 6

**Review:**

After rebuttal,

I think the authors made a valid argument to address my concerns on evaluation. So, I'd like to increase my score as weak accept!

=====


Summary:

To handle noisy labels, this paper proposed a curriculum loss that corresponds to the upper bound of 0-1 loss. Using synthetic noisy labels on MNIST and CIFAR, the authors verified that the proposed method can significantly improve the robustness against noisy labels.

Detailed comments:

Overall, the paper is well-written and the ideas are novel. However, experiments are a little weak due to weak baselines and experimental setups (see suggestions for more details). I will consider raising my score according to the rebuttal.

Suggestions:

1. Could the authors consider more baselines like D2L [Ma' 18] and Reweight [Ren' 18]

2. Similar to [Lee' 19], could the authors evaluate the performance of the proposed methods on more realistic noisy labels such as semantic noisy labels and open-set noisy labels?

[Lee' 19] robust inference via generative classifiers for handling noisy labels, In ICML, 2019.

[Ma' 18] Dimensionality-Driven Learning with Noisy Labels, In ICML, 2018.

[Ren' 18] Learning to Reweight Examples for Robust Deep Learning, In ICML, 2018.

**Experience Assessment:**

I have published one or two papers in this area.

**Review Assessment: Checking Correctness Of Derivations And Theory:**

I did not assess the derivations or theory.

**Review Assessment: Checking Correctness Of Experiments:**

I carefully checked the experiments.

**Review Assessment: Thoroughness In Paper Reading:**

I read the paper thoroughly.

---

> ### Author Response · Authors · 2019-11-15
> **More experiments for evaluation**
>
>
>
> Thanks for your comments.
>
> We provide the suggested experiments in Appendix B.
> We use the open-sourced code of Lee et al., ICML 2019 on GitHub. We only change the loss by our CL and NPCL. The experimental results show the effectiveness of our CL/NPCL. Note that CL/NPCL is a single loss for network training; one can combine them with the ensemble method (Lee et al. 2019) to boost the performance.

---

### Official Review · AnonReviewer1 · 2019-10-23
**Official Blind Review #1**

**Rating:** 6

**Review:**

This paper tackles the problem of learning with noisy labels and proposes a novel cost function that integrates the idea of curriculum learning with the robustness of 0/1 loss. The resulting cost function is theoretically justified and experimentally validated.

Pros:
(1) The proposed cost function is novel in its design, especially the aspect of curriculum learning with a computationally efficient implementation, as in Algorithm 1.
(2) The new cost function could be treated as a simple-to-implement add-on to make learning more robust to noisy labels.
(3) The introduction is well formulated and organized with focused motivation.

Cons:
(1) Equ. 9 requires more explanation of the intuition of using a combination of conventional surrogate loss and 0/1 loss, and furthermore the role of the index indicator in balancing the above two parts.
(2) Curriculum learning focuses on easy example followed by hard ones. Yet noisy examples are mixed with difficult ones in your formulation of sample selection mechanism (index indicator). The pruned examples are therefore more likely to have a high proportion of hard examples, which is undesirable. To illustrates the effectiveness of the proposed algorithm against such scenarios , one would like to see experiments on more difficult datasets such as Tiny-ImageNet.
(3) It is not clear if the quantitative results in Table 2 and 3 are produced with the pre-defined \epsilon beforehand or with grid search as done in Table 4. Knowing \epsilon would render comparison unfair for baselines.

Other remarks:
(1) E(u) threshold parameter changes from “n” in equation 11 to “C” in equation 13 (probably considering equation 9). In Equ 13, C is given as "n+0/1 loss", its transition to the other alternative forms in Equ 18 is not fully explained.
(2) The purpose of proposition 1 is unclear and may be at least shortened.
(3) Should have used some uncertainty metric instead.
(4) Incremental improvement over SOTA. SOTA was actually better in some cases.

**Experience Assessment:**

I have read many papers in this area.

**Review Assessment: Checking Correctness Of Derivations And Theory:**

I assessed the sensibility of the derivations and theory.

**Review Assessment: Checking Correctness Of Experiments:**

I carefully checked the experiments.

**Review Assessment: Thoroughness In Paper Reading:**

I read the paper thoroughly.

---

> ### Author Response · Authors · 2019-11-15
> **Explanation and more experiments.**
>
>
> Thanks for your comments.
>
> 1. Explanation of Eq.(9)
>
> The difficulty of optimizing the 0-1 loss is that the 0-1 loss has zero gradients in almost everywhere (except at the breaking point). This issue prevents us from using first-order methods to optimize the 0-1 loss. Eq.(9) provides a surrogate of the 0-1 loss with non-zero subgradient for optimization, while preserving robust properties of the 0-1 loss. Note that our goal is to construct a tight upper bound of the 0-1 loss while preserving informative (sub)gradients. Eq.(9) balances the 0-1 loss and conventional surrogate by selecting (the trust) samples (index) for training progressively.
>
> 2. We provide experiments on Tiny-ImageNet in Appendix B. We use ResNet18 as the testbed. Both symmetric 20% and symmetric 50% noise cases are evaluated. The experimental results show that NPCL can improve performance on the more difficult dataset Tiny-ImageNet.
>
> 3. In Table 2 and 3, the NPCL, Co-teaching, and Co-teaching+ use the true noise rate. We further evaluate CL on CIFAR10, CIFAR100, and Tiny-ImageNet. The experimental results are provided in Appendix B.  It shows that CL can obtain comparable results with NPCL. Moreover, CL obtains better performance on CIFAR10 and CIFAR100 with uniform noise, and competitive performance on cases with semantic noise, compared to the ensemble methods (RoG) of Lee et al., ICML 2019. Note that CL does not have parameters. It is much more convenient to use.

---

### Decision · Program_Chairs · 2019-12-19

**Decision:**

Accept (Poster)

**Comment:**

This paper studies learning with noisy labels by integrating the idea of curriculum learning.

All reviewers and AC are happy with novelty, clear write-up and experimental results.

I recommend acceptance.